# Modelling biogeochemical processes in sediments from the north western Adriatic Sea: response to enhanced particulate organic carbon fluxes

Daniele Brigolin [1], Christophe Rabouille [2], Bruno Bombled [2], Silvia Colla [1], Salvatrice Vizzini [3], Roberto Pastres [1], Fabio Pranovi [1]

[1]Department of Environmental Sciences, Informatics and Statistics, Università Ca' Foscari, Venezia, 30172, Italy
[2]Laboratoire des Sciences du Climat et de l'Environnement, UMR CEA-CNRS-UVSQ et IPSL, Gif sur Yvette, F-91198, France
[3]Department of Earth and Marine Sciences, Università degli studi di Palermo, Palermo, 90123, Italy

*Correspondence to*: Daniele Brigolin (brigo@unive.it)

**Abstract.** This work presents the result of a study carried out in the north-western Adriatic Sea, by combining two different types of biogeochemical models with field sampling efforts. A longline mussel farm was taken as a local source of perturbation to the natural Particulate Organic Carbon (POC) downward flux. This flux was first quantified by means of a pelagic model of POC deposition coupled to sediment traps data, and its effects on sediment bioirrigation capacity and OM degradation pathways were investigated constraining an early diagenesis model by using original data collected in sediment porewaters. The measurements were performed at stations located inside and outside the area affected by mussel farm deposition. Model-predicted POC fluxes showed marked spatial and temporal variability, which were mostly associated with the dynamics of the farming cycle. Sediment traps data at the two sampled stations (in and out of the mussel farm) showed average POC background flux of 20.0-24.2 mmol C $m^{-2}$ $d^{-1}$. The difference of OC fluxes between the two stations was in agreement with model results, ranging between 3.3 and 14.2 mmol C $m^{-2}$ $d^{-1}$, and primarily associated with mussel physiological conditions. Although restricted, these changes in POC fluxes induced visible effects on sediment biogeochemistry. Observed oxygen microprofiles presented a 50 % decrease in oxygen penetration depth (from 2.3 to 1.4 mm), accompanied by an increase in the $O_2$ influx at the station below the mussel farm (19-31 versus 10-12 mmol $O_2$ $m^{-2}$ $d^{-1}$) characterized by higher POC flux. DIC and $NH_4^+$ concentrations had similar behavior, with a more evident effect of bioirrigation underneath the farm. This was confirmed through constraining the early diagenesis model, which calibration leads to an estimation of enhanced and shallower bioirrigation underneath the farm: bioirrigation rates of 40 $y^{-1}$ and irrigation depth of 15 cm were estimated inside the shellfish deposition footprint versus 20 $y^{-1}$ and 20 cm outside. These findings were confirmed by independent data on macrofauna composition collected at the study site. Early diagenesis model results indicated a larger organic matter mineralization below the mussel farm (11.1 versus 18.7 mmol $m^{-2}$ $d^{-1}$), characterized by similar proportions between oxic and anoxic degradation rates at the two stations, with an increase in the absolute values of oxygen consumed by OM degradation and reduced substances re-oxidation underneath the mussel farm.

## 1 Introduction

Disturbance gradients in benthic habitats of marine soft sediment can cause shifts in species behaviours and interactions, thus affecting the biodiversity-ecosystem function relationship (Snelgrove et al., 2014; Villnäs et al., 2013). Disturbance gradients can also affect spatial habitat heterogeneity, determining changes in essential ecosystem traits such as food web functioning (Rooney et al., 2008). Mathematical models of early diagenesis - reaction transport models allowing to simulate organic matter degradation, kinetic and equilibrium reactions, as well as biologically mediated processes, such as bioturbation and bioirrigation (e.g. Arndt et al., 2013) - can represent useful tools to study sediment biogeochemistry and animal-sediment interactions under specific conditions of organic matter deposition, and to generalize results from in-situ observation and/or

lab experiments (Volkenborn et al., 2012). However, their use for studying the effects of disturbance gradients in coastal sediments was very limited in the past (Paraska et al., 2014).

Shellfish farming is regarded as an extractive aquaculture activity (Barrington et al., 2009). However, the production of faeces and pseudofaeces (excess particles rejected by palps before ingestion) leads to a net transfer of organic matter from the water column to the surface sediment (Tenore and Dunstan, 1973; Cranford et al., 2007). This process is expected to locally affect sediment biogeochemistry, benthic-pelagic coupling and benthic community functioning. A range of studies peformed in the last 30 years reported on farm-induced changes in sedimentation rates (Callier et al., 2006), sulfate reduction (Dahlbäck and Gunnarson, 1981), $NH_4^+$ and $PO_4^{2-}$ regeneration (Hatcher et al., 1994; Nizzoli et al., 2005), porewater nutrient concentration gradients (Mesnage et al., 2007), benthic community structure (Stenon-Dozey et al., 1999; Mirto et al., 2000; Christensen et al., 2003). Only recently, the quantitative understanding of these processes has received attention in relation to the assessment of "positive modifications" induced by shellfish farm deposition on benthic habitats (McKindsey et al., 2011). The influence of local hydrodynamics on the fluxes of organic matter deposited by mussel farms was the focus of two modelling studies, by Hartstein and Stevens (2005) and Weise et al. (2009). Based on these works it was possible to have a clearer mechanistic understanding of the relationship between the values of flux and the area affected by organic deposition and the different farming conditions (in terms of local hydrodynamics and farm characteristics – depth; geometry). To our knowledge, less attention was received by the pelagic-benthic coupling associated to the different phases of the rearing cycle, and, ultimately linked to the physiology of the farmed mussel - faeces and pseudofaeces production rates are dependent on water temperature and seston quantity and quality (Tenore and Dunstan, 1973), parameters which could present remarkable variations at the annual time scale, in particular at those sites characterized by a relatively fast growth out cycle – e.g. the Mediterranean Sea.

In this work, a longline mussel farm located in the north western Adriatic Sea was regarded as a local source of perturbation of natural organic matter downward fluxes. Average yearly increase in Particulate Organic Carbon (POC) flux induced by the mussel farm throughout the year was first quantified by applying a biogeochemical model of POC production and deposition (mussel faeces and pseudofaeces), coupled to two sediment trap deployments, which were carried out at the beginning and at the end of mussels farming cycles, with the aim of corroborating model predictions. Outputs of this first model, were subsequently used in early-diagenesis model simulations (one steady-state and one transient), which were constrained by the observed field data in the sampled cores at two stations (pristine and impacted): bioirrigation parameters and ratio among degradation pathways were estimated on the basis of model application. Measurements included $O_2$ micro-profiling, porosity and micro-porosity, pore waters $NH_4^+$, $SO_4^{2-}$, and Dissolved Inorganic Carbon (DIC). The objectives of this study are a better understanding of seasonal and inter-annual variability in POC deposition fluxes from the mussel farm, and the quantification of benthic recycling of organic matter under contrasted forcings linked to mussel farm, ecological mechanisms associated (bioirrigation), and relative importance of oxic versus anoxic processes.

## 2 Materials and methods

### 2.1 Study site and mussel farm description

The study was performed at a Mediterranean mussel (*Mytilus galloprovincialis*) farm located approximately 1.5 nautical miles off-shore the city of Jesolo (Italy), in the North-Western Adriatic Sea (Figure 1). The area is characterized by a flat bathymetry ranging between 13 and 15 m (Colla, 2017). The farm is affected by the freshwater plume of the Sile River, which outlet is located at approximately 1.5 nm of distance from the south-western edge of the farm (Figure 1). The mean annual Sile river discharge in 2008-2009 was 10.9 $m^3 s^{-1}$, ranging between 5.2 $m^3 s^{-1}$ in March 2008, and 14.7 $m^3 s^{-1}$ in December 2009 (ARPAV, 2010). A grain size analysis was recently carried out in the area: sediments were classified as silty-sandy, with percentages of sand and mud respectively within the ranges 11.9-25.5 % and 88.1-74.5 % (Colla, 2017). Offshore mussel farming is extensively practiced in the northern Adriatic Sea, and farmed mussels account for approximately 2/3 of the national Italian

production (MiPAAF, 2014). The studied mussel farm covers an area of about 2 km$^2$ and has been operating since 1990 with an average annual production ranging between 600 and 800 tons year$^{-1}$ (farm manager pers. comm.). Mussels are grown on ropes approximately 4 m long, which are suspended on cables, and placed at depths comprised between 2 and 4 m. Lines are positioned parallel to the coast, along the principal current direction, at a distance of 40 m among each other. Length of each line is approximately 2 km. Mussels are normally harvested within July-September, after a rearing cycle lasting a single year, during which they are re-socked 2-3 times (farm manager pers. comm.).

## 2.2 Model theory: mussel farm deposition model and Early diagenesis model

For the purposes of this work, the model described by Brigolin et al. (2014) was modified to simulate the C, N, P biogeochemical fluxes across shellfish farms. For an introductive description of the model, the reader should refer to the original publication, here we will focus on those changes required to adapt the model to the simulation of mussel farms deposition. The model (Figure 2) combines two generic modules, respectively accounting for: i) individual growth and dynamics of the farmed population; ii) organic particle tracking and deposition.

Mussel growth and population dynamics were estimated by means of the individual-based approach by Brigolin et al. (2009). This individual model is capable to simulate physiological processes and their response to key environmental forcings, i.e. suspended particulate matter quality/quantity and water temperature. The individual growth model requires in input time series of daily values of sea water temperature, and concentrations of chlorophyll-a, POC, and total suspended solids. The model allows accounting for the daily energy intake, weight gain, and faeces and pseudofaeces production rate, these latter two representing the input for the deposition module. The individual model was up-scaled to the population level by means of a set of Monte Carlo simulations, which were used for estimating the size structure of the population (the virtual population was made up of 5000 individuals). In accordance with Bacher and Gangnery (2006) such differences were accounted for by assigning to each specimen a different maximum clearance rate, reflecting variability in individual phenotypes as well as differences in the localization of specimens within the farm. The model allows to specify farm geometry based on a set of basic parameters. An interval is selected in order to account for the depth at which ropes are positioned, $z_{fmin}$-$z_{fmax}$, and initial position of mussel biodeposits (faeces and pseudofaeces) is assigned randomly within this interval. Longlines - cables on which ropes containing mussels are attached - are disposed parallel each to the other, at a fixed distance, $S_x$, and with a defined orientation with respect to the North, $D_n$. Mussels are considered to have an homogeneous density, $B_i$, within the same i-th longline. A fixed number of longlines, $n_L$, of length $l_L$, are assumed to be productive within a farming cycle. The bathymetry of the farmed area can either be specified through an external file, or assumed to be flat. Faeces and pseudofaeces deposition from mussel lines is simulated by means of a Lagrangian technique, consistent with the advection-diffusion equation (details inJusup et al., 2007). Effect of structures on the current within the farm was not accounted in the present work, due to the lack of 2D current meter data required (e.g. the work by Hartstein and Stevens (2005) for New Zealand type long-lines). This model requires in input time series of water velocity and fluxes of faeces and pseudofaeces. The latter two time series are provided by the individual-based population dynamic model with a daily resolution. Fluxes associated with the metabolic activities of the entire farmed population are computed by integrating individual fluxes over the size structure of the population, and over time, Eq. (4) in Brigolin et al. (2014). These fluxes are equally partitioned among C, N and P content to each generic particle. Particles are released homogeneously within the 24 hours, with a total of 5000 particles launched every day from each mussel line. The output of this module are daily maps of downward fluxes of organic C, N and P reaching the sea bed (in g m$^{-2}$ d$^{-1}$). The complete set of parameters used in the deposition model, values and their references, are reported in Table A1 (Appendix). Settling velocities for mussel faeces and pseudofaeces were set to 1.0 ± 0.1 cm s$^{-1}$ and 0.1 ± 0.01 cm s$^{-1}$ (Weise et al., 2009; Chamberlain, 2002). The settling velocity of each particle was randomly selected from a Gaussian distribution.

The early diagenesis model (EDM) was developed by means of the Biogeochemical Reaction Network Simulator - BRNS - (Regnier et al., 2002), through the Knowledge-Based Reactive Transport Model application (Aguilera et al., 2005). In the

present application we used a simplified version of the EDM identified for the northern Adriatic Sea by Brigolin et al. (2011). The model solves the diagenetic equations describing mass conservation for solids and dissolved species in a vertical sediment column - Eqs. (1) and (2) in Table A2 (Berner, 1980; Boudreau, 1997). The advection term includes burial, compaction, and bioirrigation; the diffusion term includes molecular and ionic diffusion, as well as bioturbation. Non-local mixing due to bioirrigation has been previously included in models developed using the BRNS (e.g. Dale et al., 2008), and in the present application we assumed a profile for bioirrigation rate of the type reported by Canavan et al. (2006) for a coastal lake (formulation reported in Table 2). Organic matter oxidation is described by means of a multi-G model (Westrich and Berner, 1984) with three types of organic matter: labile (OM1), semi-refractory (OM2), and mussel biodeposits (faeces + pseudofaeces) (OM3), following the approach by Van Cappellen and Wang (1996). Oxic and anoxic pathways of organic matter oxidation, as well as secondary redox reactions, are included (Table A2). As said, the reaction network is simpler with respect to the one published in Brigolin et al. (2011), not including reactions involving Fe and Mn, which processes were not the focus of the present work. According to Burdige (2006), in shallow depth environments, Fe and Mn contribute on average for 10% of the total mineralization, with peaks around 20% - this latter value is also in agreement with the estimations by Van Cappellen and Wang (1996). A fixed concentration was imposed at the upper boundary for all solutes, while a fixed flux was used for solids. The model was coded in Fortran. The ordinary differential equations were integrated numerically by means of a 4th order Runge-Kutta scheme (Press et al., 1987). The Lagrangian equation for the deposition model was solved following Jusup et al. (2007). The set of partial differential equations in the EDM was numerically solved by means of an implicit method; details on operator splitting technique – sequential non-iterative approach, and definition of the function residuals and the Jacobian matrix are provided in Aguilera et al. (2005) and Regnier et al. (2002). Model runs were performed on SCSCF (www.dais.unive.it/scscf), a multiprocessor cluster system owned by Ca' Foscari University of Venice running under GNU/Linux.

## 2.3 Model application: simulations set up and EDM calibration

Based on the rearing cycle characteristics, described in section 2.1, in model simulations shellfish are stocked in September, and harvested after one year. The farm, made up of $n_L$=25 longlines of lenght $l_L$=2000 m each, occupies a total area of 2 x 1 km². Longlines orientation with respect to the North, $D_n$, was of 60°. A biomass density, $B_i = 15$ ind. m$^{-2}$ was considered at all the longlines of the farm. A depth range, $z_{fmin}$-$z_{fmax}$, comprised between 1 m and 7 m was selected as representative for ropes at the farm considered in this study. The model application was aimed at constraining the typical variability of mussel deposition at the farm site. In order to simulate the average flux of POC deposited by the farm, we carried out 10 different runs, considering each one a rearing cycle under forcings for a different year within the 2002-2011 time frame. Satellite data were used as inputs for individual-based population dynamics model, in accordance with previous studies (e.g. Thomas et al., 2011; Filgueira et al., 2013). The median daily value of faeces and pseudofaeces fluxes from the 10 simulations was used as an input for the deposition model. In computing this statistics we followed an approach similar to the one reported by Sarà et al. (2013), in order to smooth potential biases introduced in the model through the forcing data. This precaution was adopted since Northern Adriatic optically shallow waters are influenced by river discharge, and chlorophyll-a concentrations could be potentially over-estimated in specific months due to the interference caused by Colored Dissolved Organic Matter absorbance (Cannizzaro and Carder, 2006). Our choice to rely on satellite-derived chlorophyll-a concentrations was supported by two main considerations:

- the analysis by Mauri et al. (2007) reported very weak correlations between Po River discharge and interpolated satellite derived chlorophyll-a concentrations in the area interested by this study, and the Sile river does not provide perturbation of water clarity except during major floods (average runoff of the Sile river is approximately 1 order of magnitude lower than Adige and Brenta, and 2 orders lower than Po (Cozzi and Giani, 2011));

- the median values of chlorophyll-a concentrations obtained from satellite data (see description below, and Figure A1) were compatible with the median values reported by Solidoro et al. (2009) based on the analysis of a 20 years of data (1986-2006) of in-situ chlorophyll-a measurements, on a portion of sea which included our study area (referred as sector 3 in their work), ranging between 1.35 and 2.38 μg L$^{-1}$ in the upper layer (0-7.5 m).

Time series of monthly sea surface temperature and concentration of chlorophyll-a were extracted from the EMIS (http://emis.jrc.ec.europa.eu/) data base from July 2002 to December 2012, (longmin 12.5; longmax 12.6; latmin 45.4; latmax 45.5) by means of the R package EMISR v0.1 (R version 3.0.3). Chlorophyll-a concentrations and sea surface temperature data were derived from the sensor Modis (Moderate Resolution Imaging Spectroradiometer) Aqua and Terra respectively, with a spatial resolution of 4km. Being the farm located at the intersection of 4 pixels, in an area characterized by a flat bathymetry,

it was considered more representative to take the concentration as an average among the 4 pixels, rather than selecting a single value. Due to the lack of long term time series of data, an average POC concentration had to be imposed, 0.1 mg L$^{-1}$, on the basis of a time series of monthly data collected at a nearby farm between 2006 and 2007 (Brigolin et al., 2009). The Particulate Organic Matter / Total Suspended Solids ratio was fixed on the basis of the time series collected within the same work (Brigolin et al., 2009), providing an average Absorption Efficiency of 0.6.

Modelling deposition requires as input time series of water velocity at an hourly time step. These data were provided on the basis of a current meter deployment carried out between March and September 2010 at a station located approximately 500 m from the NE edge of the farm (Boldrin A. pers. comm., Fig. A2). Current meter data were first processed by means of a classical harmonic analysis, in order to extract tidal components as well as long-term residual means (Pawlowicz et al., 2002). On the basis of the procedure proposed by Jusup et al. (2007), the residual currents were therefore edited randomly for short

periods of time in order to reproduce the variability recorded from current meter measurements during extreme events (i.e. storms). Number of events was imposed on the basis of the 2010 current time series, and of previous current meter deployments available for this area (Rampazzo et al., 2013; Giovanardi et al., 2003). Effects of tide and storm events were therefore accounted in the final time series, while short-period fluctuations related to turbulence were accounted for by the deposition model, as reported by Jusup et al. (2007).

A steady-state EDM simulation was carried out at station EST1 located out of the mussel farm. The model was applied by using the same parameterization adopted for the North Western Adriatic shelf by Brigolin et al. (2011) - Tables A3-A4 - and by calibrating the POC downward flux, $\Phi_{OM1+OM2}$ and the parameters, $\alpha_0$ and $x_{irr1}$, defining respectively intensity and depth of bioirrigation. Initial values of POC downward flux for the calibration were set on the basis of the results of the sediment trap experiment (see section 2.4 below). This steady-state model calibration was carried out by fitting the $O_2$, DIC, $NH_4^+$ and $SO_4^{2-}$

profiles observed at station EST1. The transient simulation carried out at station IN1 was performed by imposing as initial conditions model outputs obtained at station EST1. The model, which had the same structure of the EDM run at EST1, was run for 20 years (time of activity of the farm) by taking into account the average $OM_3$ flux (faeces+pseudofaeces) predicted by the deposition model at station IN1. $\alpha_0$ and $x_{irr1}$ parameters were calibrated by fitting the $O_2$, DIC, $NH_4^+$ and $SO_4^{2-}$ profiles observed at station IN1. Diffusive oxygen uptake was calculated from profiles (both model and data, see section 2.5 below)

by means of the 1-D Fick's first law of diffusion. These fluxes were assessed from oxygen profiles by considering the oxygen gradient within the thin diffusive boundary layer. Temperature and salinity corrections were accounted for, based on measurements performed on bottom water samples. Porosity was taken into account, and the calculation of the diffusion coefficients was done in accordance with Andrews and Benett (1981).

**2.4 Sediment traps measurements**

As part of this study, sedimentation fluxes were measured by means of sediment traps positioned at the bottom. Two 48 hours sediment trap deployments were carried out. The first experiment was performed between 29/08/2014 and 31/08/2014, at the end of the annual mussel rearing cycle. The second, was carried out between 11/09/2015 and 13/09/2015, at the very beginning

of the annual mussel rearing cycle. Three PVC (polyvinyl chloride) sediment traps were deployed at each station: cylindrical shape; aspect ratio 5:1; collecting area of 0.0095 $m^2$ each (Cromey et al., 2002; Jusup et al., 2009). In the first experiment (Figure 1), sediment traps were deployed at four locations; two stations IN1 and IN2 located inside the modeled depositional footprint, and two stations outside EST1 and EST2. For the second experiment, one station was located inside IN1 and one outside EST1. Upon collection traps content was filtered through pre-combusted (450°C, 4h) and pre-weighed Whatmann GF/F filters. For total mass flux determination, filters were dried at 60 ∘C for 24 h and re-weighed. For POC determination, filters were stored at -20°C until analysis, which was carried out by means of a Thermo Elementar Analyzer (Flash - EA 1112), after acidification with HCl for removing carbonates. The percentage of organic carbon on total mass (OC%) was calculated from POC fluxes and total mass fluxes.

## 2.5 Sediment coring, microelectrode measurements, porewater and solid phase analyses

Sediment were sampled at stations IN1 and EST1 in June 2015 (respectively on 23/06 and 24/06). Undisturbed cores were collected by means of an Uwitec corer (10 cm diameter; 20 cm avg. penetration depth). Water was sampled 2 m above the bottom by means of a Niskin bottle, for dissolved oxygen, salinity and temperature determinations. Cores were immediately brought back to the field camp and prepared for microprofiling, which was conducted a few hours after coring. As the temperature of the outside air was within a few degrees of the water temperature during the cloudy sampling days (23°C in air versus 21°C in the water), the temperature was not controlled using the available cryostat, but monitored at the start and end of the measurements, and showed minimal variations. Cores were bubbled with air during measurements to allow aeration and gentle stirring. As the bottom waters were saturated with oxygen, bubbling maintained the proper in situ $O_2$ conditions. Microprofiling was conducted with a Unisense motorized microprofiler. Four oxygen microprofiles were performed using 100 μm tip microsensors which were calibrated by a two-points method: Winkler titration of the overlying water (with a precision of 2 permil) and zero-oxygen signal in the anoxic layer below the oxic zone. Porosity was obtained by measuring the weight loss upon drying at 60°C, until constant weight. Porosity was recalculated from this weight loss using salt correction and dry bulk density. Porewaters were extracted within 4 hours after coring in a glove bag under $N_2$ using Rhizons® (Seeberg-Elverfeldt et al., 2005). Samples were preserved using $HgCl_2$ saturated for DIC and Total Alkalinity (TA) analysis, by freezing for $NH_4^+$ determination and by acidifying for $SO_4^{2-}$ analysis. Measurements were performed in the laboratory: DIC was analysed on a DIC analyzer (Apollo SciTech®) using 1mL sample volume with 4 to 6 replicates which provided a standard deviation of 0.5%. TA concentrations were measured in a potentiometric opencell titration on 3mL sample volume (Rassmann et al., 2016) with an uncertainty of 0.5%. Ammonium concentration was measured spectrophotometrically following Grasshof et al. (1983) with an uncertainty of about 5%. Sulfate were measured by HPLC (High-performance liquid chromatography) on a Dionex Ionic Chromatography System (ICS 1000) using a Dionex IonPac AS9-HC Carbonate Eluent Anion-Exchange Column (4 x 50 mm) and a Dionex IonPac AG9-HC Guard Column (4 x 50 mm) with an uncertainty of 1%.

## 3 Results

### 3.1 POC reaching the sediment-water interface: model simulation results and sediment trap experiments

Results obtained by means of the population dynamic model are reported in Figures 3a-d. The time series of satellite data, sea surface temperature and chlorophyll-a concentration, used to force the model are shown in Figure A1. Growth trajectories, here expressed in terms of soft tissues dry weight and shell length are in agreement with previous observations and model results obtained for this area (Brigolin et al., 2009). The minimum commercial size of 5 cm is achieved rapidly (<6 months), and the mussels reach the final length of 7 cm within 10-11 months. At the end of the cycle mussels present weights comprised between 1.5 and 2.5 g. Total faeces and pseudofaeces release rates per line are shown in Figure 3c,d. These fluxes are highly

variable, both at the seasonal (average coefficient of variation (CV) within-year for faeces=0.67; pseudofaeces=1.47) and inter-annual time scales (average among years CV faeces=0.30; pseudofaeces=1.46).

Figure 4 shows the map of the model-predicted fluxes of organic C induced by the mussels reaching the sediment at days 10, 120, 240 and 360 of simulation (Sep 10; Dec 29; Apr 28; Aug 26). As can be seen, magnitude of the fluxes increases with mussel growth. Maximum organic carbon fluxes predicted at day 10 and 360, representative of the situation at the beginning and at the end of the growth-out cycle, are 2.5 and 13.3 mmol C m$^{-2}$ d$^{-1}$, respectively. Footprint induced by the presence of the lines is clearly visible at days 120 and 360. At day 240, a remarkable displacement of deposition towards SW (approximately 200 m) was detected. In the other cases (Figs. 4a,b,d), the maximum footprint distance from the edge of the farm is 50m.

Total mass fluxes, POC fluxes, and OC%, measured in August 2014 and September 2015 are reported in Table 1. In August 2014, POC fluxes at the end of the rearing cycle showed significantly higher values in stations IN1 and IN2 than in EST1 and EST2 (Mann–Whitney one-tailed; n1=n2=6; p=0.03), while differences among IN1 and EST1 detected in September 2015 at the beginning of the rearing cycle were on the order of 3.3 mmol C m$^{-2}$ d$^{-1}$, and not significant (Mann–Whitney two-tailed; n1=n2=3; p>0.5).

### 3.2 Early diagenesis processes underneath the farm and at a nearby station located outside the farm influence

Bottom water temperature and salinity were respectively 22 °C and 36.3 psu at both stations. Oxygen in bottom waters, measured through Winkler titration, was $223.5 \pm 0.3$ μM at EST1 and $229.8 \pm 0.1$ μM at IN1. The grain size analysis classified sediments as medium silt at EST1 and very fine sandy-coarse silt at IN1. Porosity profiles (Figure 5a) show a decreasing trend going down-core, with a steep gradient in the upper 20 mm. A discontinuity is visible in IN1 core, between 40 and 50 mm. A total of 7 oxygen profiles were collected at the two stations (Figure 5b). Oxygen shows a quasi-monotonous decrease in concentration downward in the sediment. A slight increase at interface (~ 20 μM), most probably due to microphytobenthic production, is visible at station EST1 (bubbles were visible on the core surface after long-term exposure to sunlight). Indeed, results obtained at the two stations suggested a limited variability in the oxygen behavior within the same core - profiles were measured by sampling randomly the available portion of the core in which no shell debris and macrobenthos were visible at the surface. Oxygen is consumed within the first mm, showing a higher penetration depth at EST1 (2.3 mm on average) with respect to IN1 (1.4 mm on average). DIC, NH$_4^+$ and SO$_4^{2-}$ data are shown in Figures 5c-e. DIC concentration profiles at the two stations are comparable, although at depths >10 cm they stabilize at values around 3.6 mM at EST1 and 2.8 mM at IN1. A similar pattern is visible for ammonium, which average concentration below 10 cm depth reaches values of 50 μM at IN1 and 125 μM at EST1. The effect of bioirrigation is visible within the upper 10 cm, and, is particularly marked at station IN1, where DIC shows a decrease in concentration starting from 4 cm depth. SO$_4^{2-}$ concentrations are similar between the two stations, and do not present marked variations going down-core.

Model-predicted profiles (EDM) calibration are given in Figure 6, and compared with the measured O$_2$, DIC, NH$_4^+$ and SO$_4^{2-}$ field data. Porosity parameters used in the model, and given in Table 2, were estimated by independently fitting the two porosity profiles shown in Figure 5a. A metabolisable OC flux of 11.6 mmol C m$^{-2}$ d$^{-1}$ was estimated by calibrating the model at steady-state at station EST1. At IN1, an additional flux of labile organic matter (8.2 mmol C m$^{-2}$ d$^{-1}$) was imposed for 20 years, based on the median value predicted for POC flux by the pelagic deposition model. In the EDM, DIC and NH$_4^+$ profiles are both characterized by a concentration enhancement within the first cm, controlled by the degradation of the labile organic matter (OM1 and OM3), and subsequently modulated by the action of bioirrigation, which causes a down-core decrease. In general, simulated profiles reasonably agree with observed concentration values, although differences between model and data vertical trends are visible, in particular at station EST1, were predicted NH$_4^+$ exceeds observed concentrations. The depth distributions of bioirrigation coefficients were estimated independently at the two stations, obtaining values reported in Table 2, which indicate that infauna activity is higher and more concentrated in the superficial layer at IN1 than at EST1. Figure 7 compares these fluxes to the ones estimated by means of microelectrode profiles at the same stations. As can be seen, model

results are in good agreement with the ones calculated from micro-electrode profiles. Oxygen fluxes predicted by the model profiles are, respectively, 11 and 18 mmol $O_2$ m$^{-2}$ d$^{-1}$ at EST1 and IN1, while micro-electrode ones range between 10 and 12 mmol $O_2$ m$^{-2}$ d$^{-1}$, at EST1, and 19 and 31 at IN1.

Total mineralization calculated by the model is 11.1 mmol C m$^{-2}$ d$^{-1}$ at EST1, and 18.7 mmol C m$^{-2}$ d$^{-1}$ at IN1 (Table 2). Figure 8 shows the partitioning among mineralization pathways, indicated by the electron acceptors, of the total organic matter. At EST1, appoximately 27% of the total OM (3.0 mmol C m$^{-2}$ d$^{-1}$) undergoes oxic degradation, fraction similar to that observed at IN1 (30%), where the total OM flux, but also bioirrigation rate, are higher. As reported in Figure 8, 42% of the oxygen is consumed at EST1 by organic matter mineralization, and the remaining 59% is required for re-oxidizing reduced compounds (55% by reduced compounds oxidation, and 3% by nitrification). At IN1, 51% of the $O_2$ is consumed by organic matter mineralization, 8% by nitrification and 41% by reduced compounds oxidation.

## 4 Discussion

### 4.1 Constraining POC fluxes

Callier et al. (2006) measured biodeposit production from farmed *Mytilus edulis* belonging to distinct age classes (0+,1+), reporting  values of 29.1 and 44.4 mg ind.$^{-1}$ d$^{-1}$ for mussels of 4.0 and 6.9 cm respectively. Considering mussel density and farm geometry in our case study (2 km$^2$; 15 ind. m$^{-2}$; 25 lines), these values correspond to fluxes of 3.5 to 5.3 kg C line$^{-1}$ d$^{-1}$ (10 % C in biodeposits - Brigolin et al. (2009)), which are slighlty lower, but comparables, to our predictions - we found median values for faeces which range up to 10 kg C line$^{-1}$ d$^{-1}$.

The extent of the depositional area obtained in this study (on average 50 m from the edge of the farm; 14 m depth; mean current velocity of 5.4 cm s$^{-1}$) can be compared with the results obtained in previous studies. In an exposed site, Weise et al. (2009) (Cascapedia Bay, Canada; 20 m depth; mean current velocity of 10 cm s$^{-1}$), constrained the area of higher organic enrichment within 90 m from the edge of the farm. Dispersal area reported by Hatstein and Stevens (2005) was smaller, extending with a radius of approximately 30-40 m from the edge of the farm (20-30 m depth; mean current velocity of 3.4-4.0 cm s$^{-1}$). These differences in extent of the dispersal areas seem to be primarily associated to the action of currents and wave energy inducing resuspension of biodeposits accumulated on the seabed (Cromey et al., 2002). Magnitudes of OC fluxes predicted by the model were corroborated by the two sediment traps. Simulated biodeposition fluxes of 2.5 mmol C m$^{-2}$ d$^{-1}$ on September 10, at the beginning of the cycle, agree well with the very limited and not statistically valid POC flux difference (IN-EST) between average trap measurements at the same time (3.3 mmol C m$^{-2}$ d$^{-1}$). The 13.3 mmol C m$^{-2}$ d$^{-1}$ value predicted on August 26 is very close to the maximum difference between IN1 and EST1 recorded by sediment traps in August 2014, 14.2 mmol C m$^{-2}$ d$^{-1}$. These values are lower than those reported for less exposed sites (lagoons and bays). Hartstein and Stevens (2005) recorded increases in mass fluxes on the order of 60 g m$^{-2}$ day$^{-1}$ at a mussel farm in Catherine Cove and Elayne Bay (Malborough Sounds, New Zealand), and Jaramillo et al. (1992) observed differences of 153 g m$^{-2}$ d$^{-1}$ in the Queule river estuary (southern Chile). These are one to two orders of magnitude higher than the difference of 2 g m$^{-2}$ d$^{-1}$ recorded between IN1 and EST1 in August 2014. Values lower and closer to those presented here, on the order of 10 g m$^{-2}$ d$^{-1}$, were reported by Weise et al. (2009) for a longline mussel farm located at 20 m depth in a high energy environment (Cascapedia Bay, Canada). Regarding OC%, values found in the present study are in good agreement with those reported by Hartstein and Stevens (2005) (5.82-6.56 %). OC% found in the area of study (4.23-6.34 %) falls within the range reported by Giani et al. (2001) in different locations of the Northern Adriatic Sea (1.05-21.81 %). Also, background total mass fluxes measured in the present work are comparable with data measured with sediment traps by the same authors, who reported fluxes of 5.8 g m$^{-2}$ d$^{-1}$ (total mass flux) at an off-shore station located in the Northern Adriatic Sea, and mean fluxes of approximately 30.0 g m$^{-2}$ d$^{-1}$ in prodelta areas of Po and Adige rivers, with high annual variability (range 0.08 - 240 g m$^{-2}$ d$^{-1}$ ). Relatively low background values obtained in our study can be associated to lower annual discharge (10.9 m$^3$ s$^{-1}$) of Sile River than of Adige and Brenta ones (respectively, 139.5 and

85.9 m$^3$ s$^{-1}$ - 1994-2008 averages from Cozzi and Giani (2011)), being characterized by a particularly low value of the discharge rate in August and September (ARPAV, 2010) with respect to other months. Summarizing, the predicted fluxes agree well with sediment traps data, and mass fluxes measured by traps are close to values reported for similar environments, being markedly lower than those recorded in lower energy environments.

## 4.2 Influences of perturbed POC fluxes on organic matter mineralization in sediments

Absolute values of the POC fluxes obtained from the sediment trap experiments can be cross-compared to the values estimated through the inverse use of the EDM. The value of 11.6 mmol C m$^{-2}$ d$^{-1}$, at EST1 accounts for approximately 50 % of the flux measured from the traps (22 mmol C m$^{-2}$ d$^{-1}$ on average). This difference can be primarily related to the fact that the EDM accounts only for the reactive C, while the sediment trap captures all types of C, including the inert fraction. On the top of this, traps provided only a snapshot of the deposition in August and September - disregarding the seasonal variability in this deposition, and the presence of pulse deposition events. Finally, it is worth remarking that resuspension is a common mechanism in shelf trap measurements: in the northern Adriatic Giani et al. (2001) estimated the contribution of resuspension to the gross flux sedimented in traps reaching 43.7 %.

Dedieu et al. (2007) compared seasonal cycles of C, N and O inside and outside a shellfish farming area in a lagoon located in Southern France, by combining a steady state diagenetic model (Soetaert et al., 1996) with a comprehensive set of experimental data. Model results showed that organic matter accumulation at the farming area enhanced the anaerobic metabolism. Oxygen microprofiles recorded by Dedieu et al. (2007) inside and outside the mussel farm presented differences which are comparable to those recorded in the present work, with a decrease of 50% in oxygen penetration depth, and an increase up to 3 times of diffusive O$_2$ fluxes (30 mmol O$_2$ m$^{-2}$ d$^{-1}$ versus 90 mmol O$_2$ m$^{-2}$ d$^{-1}$). In Dedieu et al. (2007) this was accompanied by a remarkable enhancement in NH$_4^+$ concentrations, which was not visible in the field data reported in the present study. This can certainly be related to the difference in the rate of biodeposit accumulation, although other factors can also play a role (changes of local macrobenthic activity, and of coupled nitrification-denitrification rates). Mean OC fluxes estimated by Dedieu et al. (2007) by calibrating the diagenesis model at the mussel farm site, were 38.4, 108.0, and 108.0 mmol C m$^{-2}$ d$^{-1}$, respectively in winter, spring and summer, with increases by 26.4, 53.4, and 52.4 mmol C m$^{-2}$ d$^{-1}$ with respect to a station of reference located outside the farm influence, in the respective seasons. In our work, the background flux (OM1+OM2 flux at EST1) was lower, 11.6 mmol C m$^{-2}$ d$^{-1}$ with an increase of 8.2 mmol C m$^{-2}$ underneath the farm, at IN1. The relative increase, with respect to the background flux, was of 71 %, which is comparable to the 69 % increase found by Dedieu et al. (2007) in winter. Nonetheless, in absolute terms, Dedieu et al. (2007) reported a difference of approximately 50 mmol C m$^{-2}$ d$^{-1}$ in summer, in a system already characterized by large fluxes of organic matter, and hence dominated by SO$_4^{2-}$ reduction, whereas in this study, the increase is 5 times less - 10 mmol C m$^{-2}$ d$^{-1}$, over a system which has mostly oxic and suboxic diagenesis. Difference in absolute values of OC fluxes can be related to a higher mussel stocking density (production in the Thau lagoon is 2333 tons km$^{-2}$ versus 1450 tons km$^{-2}$ at the farm in Jesolo), lower depth (7 m Thau Lagoon, 14 m Jesolo), and hydrodynamic regime of the site (lagoon environment with low energetics in term of hydrodynamics - current meter records not available from the study). In our case, even if the relative change of POC flux between the mussel farm and the reference is large, it does not affect significantly the porewater composition at depth. We cannot rule out spatial heterogeneity within each station, which would smooth the difference between IN1 and EST1. However, the fact that O$_2$ fluxes reflect the increased input by the mussels, points towards internal mechanisms that regulate porewater composition. The shape of the DIC–NH$_4^+$ profiles indicates bio-irrigation (Meile et al, 2001; Canavan et al., 2006), although the deeper increase is not visible in our data profiles due to limited penetration of the cores. Indeed, the very limited increase in concentration profiles in the first centimeters can only be linked to input of bottom water with lower DIC and NH$_4^+$ by irrigation, given the large recycling intensity in surface sediments as exemplified by O$_2$ profile. Therefore, one way to explain the lack of differences between the IN1 and EST1 porewater composition is to relate it to changes in the bioirrigation profiles at the two stations. We remark here that $\alpha_0$ and

xirr$_1$ were the only two parameters calibrated at IN1, and they suggest a higher infauna activity, shifted towards the surface at this site. This feature was independently confirmed by a set of macrobenthos samples collected at the two stations as a part of a complementary study (Colla, 2017). Macrobenthos samples showed a higher diversity (48 vs 31 taxa recorded) and abundance (on average 1900 vs 1000 ind. m-2) at IN1 with respect to EST1, accompanied by the presence of larger organisms (0.065 g ind.$^{-1}$ at IN1 versus 0.034 g ind.$^{-1}$ at EST1). This is in agreement with the expected influence of biodeposition from mussel culture (McKindsey et al., 2011). Species recognized as important bioturbators (as *Lagis koreni, Glycera unicornis, Sipunculus nudus, Eunice vittata, Hilbigneris gracilis, Amphiura chiajei, Ensis minor, Dosinia lupinus, Tellina distorta, Nassarius incrassatus)* were present in both samples, accounting for approximately 18% of the total abundance at EST1 and for the 35% at IN1. Calibrated values of $\alpha_0$ (20 y$^{-1}$ at EST1 and 40 y$^{-1}$ at IN1) are close to those estimated by Meile et al. (2001) for Buzzards Bay site, 30-60 y$^{-1}$ (water depth 15m), and slightly higher than those estimated by Canavan et al. (2006), through data fitting - 10 y$^{-1}$ at a coastal freshwater lake (Haringvliet, in the Netherlands).

According to model estimations, total mineralization at EST1 accounts for 96 % of deposited OM (OM1+OM2), with 0.5 mmol C m$^{-2}$ d$^{-1}$ escaping mineralization through burial. This fraction decreases to 94 % at IN1 (1.1 mmol C m$^{-2}$ d$^{-1}$ of non-mineralized POC), where the model is run under transient conditions. The relative contribution of different mineralization pathways to the total OM degradation is comparable to that reported by Pastor et al. (2011) for stations with comparable OM fluxes in the Rhône River prodelta and shelf area. At stations undergoing organic carbon deposition ranging between 7.3 and 16.2 mmol C m$^{-2}$ d$^{-1}$ (their stations L, I, C, J, F), these authors found oxic mineralization ranging between 44 % and 67 %, nitrification between 1 % and 6 %, and anoxic mineralization between 27 % and 51 %. The largest fraction of oxygen, 67 %-87 % was consumed by OM degradation, nitrification consumed up to the 31 % of the O$_2$, while the re-oxidation of anaerobic products accounted for only the 2.1 % of oxygen consumption. This marks a difference with respect to the results obtained in the present work, since in our model a larger fraction of oxygen is consumed for reduced substances oxidation (54 % at EST1 and 41 % at IN1). This is also visible in Figure 6, where relevant drops in SO$_4^{2-}$ concentration are predicted by the model below the bioirrigated layer - approximately 2 mM at EST1 and 5 mM at IN1. The different behavior may be due to the processes controlling H$_2$S and Fe in Pastor et al.'s study where most H$_2$S is precipitated as FeS$_2$ thus escaping re-oxidation by sulfides. A more precise estimation of the fate of this oxygen could be obtained by introducing in the model FeS precipitation, for which at least Fe$^{2+}$ measurements in pore waters would be required.

The higher influx of oxygen and enhanced bioirrigation at the mussel site (IN1) reflected on a substantial change in the pathways of oxygen consumption, with an increase of oxic degradation of OM2, and a relative decrease of oxygen demanded for reduced substances re-oxidation -this is clearly linked to the input of fresh OM from the mussels which is mineralized aerobically. Nonetheless, due to the higher absolute fluxes of OM, the oxygen consumed by reduced substances oxidation is higher at IN1 7.4 mmol O$_2$ m$^{-2}$ d$^{-1}$, than at EST1, 6.2 mmol O$_2$ m$^{-2}$ d$^{-1}$. Higher NH$_4^+$ concentration predicted at station EST1 with respect to field data could be explained with a higher rate of nitrification at this station. However, in the calibration performed within this work, the kinetic constant for nitrification was kept at its original value (Table A4), due to the lack of data concerning NO$_3^-$. Our model results indicated that denitrification contributed between 1 and 2% to the total mineralization, which is comparable to previous estimations in the Mediterranean Sea near the Rhone River (Pastor et al., 2011; 0.1- 4%), and in the Northern Adriatic, 2-4% (Capet, unpublished data), and slightly lower than what estimated for the Black Sea (Capet et al., 2016; 5-6%).

## 4.3 Integrated model features

The pelagic deposition model allowed simulating the extent of the deposition area, and its variability with time. Being integrated with a daily time step, the Mediterranean mussel population dynamic model allows to combine instantaneously the non-linear effects of the different environmental variables and physiological processes acting on deposition (i.e. water temperature, chlorophyll-a concentration, allometric dependence of the clearance rate on body size), and integrate these effects

along the time of the farming cycle. The combination of the bioenergetics-based population model, which allows estimating organic matter production from the lines, with the deposition model, accounting for particles dispersion, represents a novel aspect of the present work with respect to previous modeling studies on mussel deposition. Hartstein and Stevens (2005) modeling study applied a sensitivity approach to study organic deposition from *Perna canaliculus* in New Zealand, comparing sites characterized by different hydrodynamic exposure, and assuming an arbitrary particle-release rate. Weise et al. (2009) modeled mussel biodeposition at different sites in the eastern coast of Canada, imposing organic wastage from the farm lines as a model input, on the basis of site-specific field measurements (Callier et al., 2006), and extrapolation from other sites. It is worth remarking here that the integration of growth and deposition models can represent a resource, allowing to apply the model at different sites in which environmental variables are known, without the need of performing in situ estimations of biodeposis production. On the top of this, the model could be used to explore the effect of climate-change-induced long term trends of variation in water temperature and particulate organic matter concentrations, which are expected to have an influence on mussel growth performances (Cochrane et al., 2009; Rosa et al., 2012). We underline that the application presented in this work could be extended, in order to include the evaluation of the uncertainties related to spatial inconsistencies of nearshore-offshore remote sensing products.

## 5 Conclusions

The combined application of an early diagenesis model and of a model of POC production and deposition from shellfish filter-feeders, allowed to study quantitatively the differences induced on sediment biogeochemistry by a local perturbation of the natural POC downward flux. This is one of the few existing attempts to couple pelagic mussel production models with early diagenetic models in order to investigate the effects of a local gradient of disturbance on coastal sediments biogeochemistry and benthic-pelagic coupling (review by Paraska et al., 2014). Model-predicted POC fluxes showed marked spatial and temporal variability. Sediment traps data at the two sampled stations were in agreement with model results. The increase of POC fluxes by 100 % caused by the mussel farm induced visible effects on sediment biogeochemistry. Measured oxygen microprofiles showed a remarkable decrease in oxygen penetration depth, approximately 50 %, accompanied by an increase in the $O_2$ influx at the station characterized by higher POC. DIC and $NH_4^+$ concentrations had similar behaviour, with a more obvious effect of bioirrigation underneath the farm. Indeed, the early diagenesis model calibration led to an estimation of enhanced and shallower bioirrigation underneath the farm which were confirmed by independent data on macrofauna composition collected at the study site. We remark that, based on the number of cores available, it was not possible to assess quantitatively the uncertainty related to these coefficients, which estimation would allow to better characterize bioirrigation in this area. Early diagenesis model results indicated a similar proportion between oxic and anoxic degradation pathways at the two stations, with an increase in the absolute values of oxygen consumed by OM degradation and reduced substances re-oxidation underneath the mussel farm. Model estimates an area of 159000 $m^2$ (approx 8 % of the farm lease) characterized by deposition fluxes >= 8 mmol C $m^{-2}$ $d^{-1}$ ( $\approx 0.1$ g C $m^{-2}$ $d^{-1}$). Enhancement of $O_2$ influx induced by the farm -with respect to a non-farmed area of the same dimension – ranges from 4.6 $10^5$ mol $O_2$ $y^{-1}$ (via EDM estimation) to 7.2 $10^5$ mol $O_2$ $y^{-1}$ (calculated from profiles). These results can help in the assessment of the role of disturbance gradients, such as an increased POC flux, in affecting sediment biogeochemical conditions and spatial habitat heterogeneity. From an applied perspective, knowledge and representation of these processes is fundamental to attempt a sound management of the marine space in those coastal ecosystems in which mussel farming is extensively practiced.

## 6 Appendix

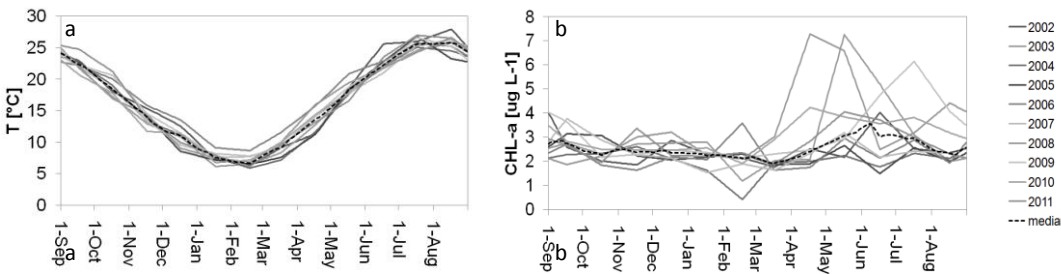

**Figure A1. model forcings: a) sea water temperature; b) chlorophyll-a concentration. Time series for years 2002-2011, and the overall medians are reported.**

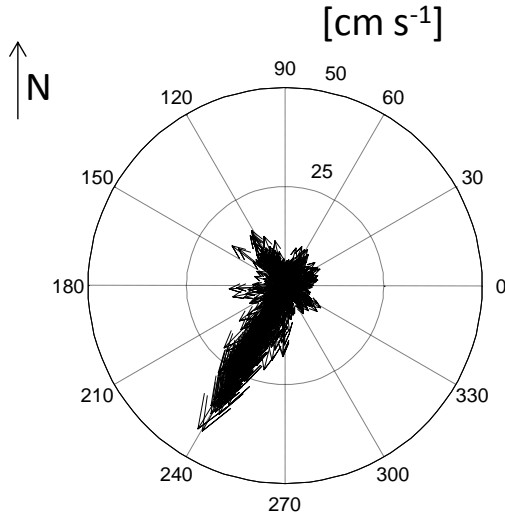

**Figure A2. Module [cm s⁻¹] and directions of the water currents as recorded nearby the farm between March and September 2010.**

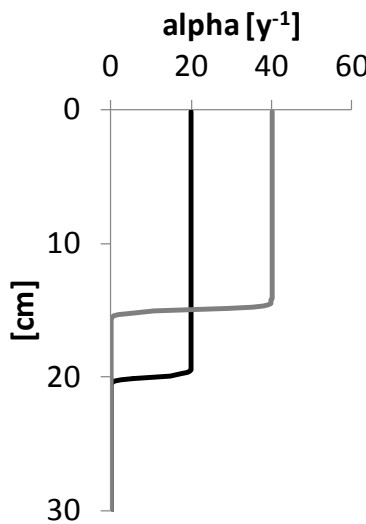

10  **Figure A3. Profiles of the irrigation rate used in the model applications: EST1 black line; IN1 gray line.**

**Table A1. Parameters used in the deposition model.**

| Name | Description | Value | Units | Reference |
|---|---|---|---|---|
| $dx, dy$ | Horizontal resolution | 20 | [m] | - |
| $dt$ | Time step | 60 | [s] | - |
| $K_x, K_y$ | Horizontal eddy diffusivity coefficient | 0.1 | [$m^2\ s^{-1}$] | Cromey et al. (2002), Jusup et al. (2007) |
| $K_z$ | Vertical eddy diffusivity coefficient | 0.001 | [$m^2\ s^{-1}$] | Cromey et al. (2002), Jusup et al. (2007) |
| $w_f$ | Normal distribution of settling velocity of faeces | $\mu=1.0$; $\sigma=0.1$ | [$cm\ s^{-1}$] | Weise et al. (2009) |
| $w_p$ | Normal distribution of settling velocity of pseudofaeces | $\mu=0.1$; $\sigma=0.01$ | [$cm\ s^{-1}$] | Weise et al. (2009) |

**Table A2. Reaction network implemented in the EDM model. Eq.s 1 and 2: $C_s$ and $C_w$ are, respectively, the concentration of solid and dissolved species, $t$ denotes the time, $z$ denotes the depth below the SWI, $\varphi$ is sediment porosity, $D$ is the total molecular diffusion plus bioturbation, $D_b$, $\omega$ is the sedimentation rate, and $\Sigma R$ represents the net rate of concentration change due to chemical and biological sources and sinks. The advection term includes burial and compaction; the diffusion term includes molecular and ionic diffusion, as well as bioturbation. The network is a simplified version of the one proposed by Van Cappellen & Wang (1996). Reactions 1–3 implemented separately for each OM fraction.**

**Reaction-transport equations:**

$$\frac{\partial\left[(1-\phi)C_s\right]}{\partial t} = -\frac{\partial\left[\omega(1-\phi)C_s\right]}{\partial z} + \frac{\partial}{\partial z}\left[D(1-\phi)\frac{\partial C_s}{\partial z}\right] + (1-\phi)\sum R \tag{1}$$

$$\frac{\partial\phi C_w}{\partial t} = -\frac{\partial\omega\phi C_w}{\partial z} + \frac{\partial}{\partial z}\left[D\phi\frac{\partial C_w}{\partial z}\right] + \phi\sum R \tag{2}$$

**Primary redox reactions**

1. Oxic respiration

$(CH_2O)_x(NH_3)_y(H_3PO_4)_z + xO_2 + (-y + 2z)HCO_3^- \xrightarrow{R_1}$

$(x - y + 2z)\, CO_2 + yNH_4^+ + zDIP + (x + 2y + 2z)\, H_2O$

2. Denitrification

$(CH_2O)_x(NH_3)_y(H_3PO_4)_z + \left(\dfrac{4x + 3y}{5}\right)NO_3^- \xrightarrow{R_2}$

$\left(\dfrac{2x + 4y}{5}\right)N_2 + \left(\dfrac{x - 3y + 10z}{5}\right)CO_2 + \left(\dfrac{4x + 3y - 10z}{5}\right)HCO_3^- + zDIP + \left(\dfrac{3x + 6y + 10z}{5}\right)H_2O$

3. Sulphate reduction

$(CH_2O)_x(NH_3)_y(H_3PO_4)_z + \left(\dfrac{x}{2}\right) \cdot SO_4^{2-} + (y - 2z)CO_2 + (y - 2z)H_2O \xrightarrow{R_3}$

$\dfrac{x}{2}TS + (x + y - 2z) \cdot HCO_3^- + yNH_4^+ + z\, DIP$

**Secondary redox reactions**

4. Nitrification

$NH_4^+ + 2O_2 + 2HCO_3^- \xrightarrow{R_4} NO_3^- + 2CO_2 + 3H_2O$

5. Sulphide oxidation by $O_2$

$TS + 2O_2 + 2HCO_3^- \xrightarrow{R_5} SO_4^{2+} + 2CO_2 + 2H_2O$

**Table A3. Rate laws used in the EDM. Rates refer to the reactions listed in Table A2 of this Appendix. $f_i$ were computed according to the formulation reported in Aguilera et al. (2005).**

**Rate laws**

$R_1 = f_{o_2} \cdot k_{OM_k} \cdot [OM_k] \cdot k_{acc}$  ,with k=1,2,3

$R_2 = f_{NO_3^-} \cdot k_{OM_k} \cdot [OM_k] \cdot k_{acc}$  ,with k=1,2,3

$R_3 = f_{SO_4} \cdot k_{OM_k} \cdot [OM_k]$  ,with k=1,2,3

$R_4 = k_4 \cdot [NH_4^+] \cdot [O_2]$

$R_5 = k_5 \cdot [TS] \cdot [O_2]$

**Table A4. Reaction specific and general parameters used in the EDM. (1=Wang & Van Cappellen, 1996; 2= Jourabchi et al., 2005; 3= Canavan et al., 2006; 4= Berg et al., 2003)**

| Parameter name | Value | Units | Description | Source |
|---|---|---|---|---|
| $O_{2\ lim}$ | $16.0 \times 10^{-6}$ | $mol\ L^{-1}$ | limiting concentration for $O_2$ | Mean value from 1,2,3,4 |
| $NO_3^-{}_{\ lim}$ | $4.7 \times 10^{-6}$ | $mol\ L^{-1}$ | limiting concentration for $NO_3^-$ | Mean value from 1,2,3,4 |
| $SO_4^{2-}{}_{\ lim}$ | $1180.0 \times 10^{-6}$ | $mol\ L^{-1}$ | limiting concentration for $SO_4^{2-}$ | Mean value from 1,2,3 |
| $k_4$ | $1.2 \times 10^7$ | $(mol\ L^{-1})^{-1}\ y^{-1}$ | kinetic constant for Nitrification | Mean value from 1,2,3,4 |
| $K_5$ | $2.7 \times 10^8$ | $(mol\ L^{-1})^{-1}\ y^{-1}$ | kinetic constant for sulphides oxidation by $O_2$ kinetic constant | Mean value from 1,2,3,4 |
| | 0.25 | $cm\ y^{-1}$ | Vertical velocity | Alvisi and Frignani (1996) |
| | $10^{-4}$ | y | Time step | - |
| | 30 | cm | Max. depth modelled | - |
| | 601 | - | Grid nodes | - |

## 7 Acknowledgements

DB was supported by an individual mobility grant from the University of Venice Ca' Foscari (IRIDE-DHAMACO project). Model runs were performed on SCSCF (www.dais.unive.it/scscf), a multiprocessor cluster system owned by Ca' Foscari University of Venice running under GNU/Linux. We gratefully acknowledge G. Monvoisin from GEOPS France for the assistance in the analyses, A. Boldrin from CNR for giving access to the current meter data, and Nautica Dal Vì for providing space for setting up lab facilities. We would like to thank the two anonymous reviewers for their constructive comments on the initial version of this manuscript.

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

**Figure Captions**

**Figure** 1**: Study site. The dotted rectangle marks the edge of the area in which the mussel farm is located. Sile river outlet is visible along the coast, northward of the farm area. Stations sampled in this study are marked as black dots (IN1, IN2: under the influence of the farm; EST1, EST2: nearby stations, outside the influence of the farm).**

**Figure 2: Information flow within this study. The typical structure of a longline mussel farm is schematically represented in the upper-left part of the figure: cables (blue lines) are attached to floating buoys (red areas), and ropes (black rectangles), in which mussels are stocked. Black particles (small circles) represent the background POC flux, while green and orange particles stand for mussel faeces and pseudofaeces. Sediment traps (in triplicate) are represented as small gray rectangles, while larger rectangles in the right end side of the figure represent the sediment cores.**

**Figure 3: a,b) Mussel soft tissues dry weight and shell length; c,d) faeces and pseudofaeces release fluxes per mussel line per day. Results of 10 different model runs and the overall medians are reported (years 2002-2011). Forcings provided in input to the model are presented in Figure A1. Discontinuities in figure a) represent weight losses due to spawning events (model theory in Brigolin et al., 2009).**

**Figure 4: Model predicted fluxes of organic C reaching the sediment at simulation days 10 (September 10), 120 (December 29), 240 (April 28) and 360 (August 26). The black dot marks the north-eastern edge of the farmed area (see farm representation in Figure 1).**

**Figure 5: Measured profiles and micro-profiles. All y-axes refer to vertical depth in the sediment, with 0 set at the sediment-water interface. a) sediment porosity [-]; b) dissolved oxygen concentration – squares; triangles, diamonds and circles represent different**

microprofiles, black color=IN1; white color=EST1; c) $NH_4^+$ concentration [μM]; c) Dissolved Inorganic C concentration (DIC) [μM]; d) $SO_4^{2-}$ concentration [mM].

**Figure 6: Early Diagenesis Model results at stations EST1 (nearby stations, outside the influence of the farm; steady-state simulation) and IN1 (under the influence of the farm; transient simulation).**

**Figure 7: Model derived (black triangles) versus micro-electrode (box plots) estimations of $O_2$ diffusive fluxes.**

**Figure 8: Model estimations: relative importance of different pathways on the total organic matter at EST1 (a,b), and IN1 (d,e,f); ratios of different redox pathways on total oxygen consumption at EST1 (c), and IN1 (g).**

**Tables**

**Table 1. Downward fluxes measured by sediment traps at stations IN1, IN2 (under the influence of the farm) and EST1, EST2 ( nearby stations, outside the influence of the farm), in August 2014 and September 2015.**

| | Total mass flux [g $m^{-2}$ $d^{-1}$] | | POC flux [mmol C $m^{-2}$ $d^{-1}$] | | OC% [%] | |
|---|---|---|---|---|---|---|
| **Experiment** | Aug. 2014 (end of rearing cycle) | Sept. 2015 (beginning of rearing cycle) | Aug. 2014 | Sept. 2015 | Aug. 2014 | Sept. 2015 |
| | | | | | | |
| IN1 | 6.5 ± 1.6 | 5.7 ± 0.6 | 34.2 ± 13.3 | 27.5 ±4.2 | 6.2 ± 1.2 | 5.9 ± 0.3 |
| EST1 | 4.6 ± 0.6 | 6.9 ± 1.4 | 20.0 ± 5.0 | 24.2 ±5.0 | 5.2 ± 0.7 | 4.2 ± 0.2 |
| IN2 | 5.9 ± 1.6 | | 32.5 ± 14.2 | | 6.3 ± 1.1 | |
| EST2 | 4.6 ± 0.7 | | 20.8 ± 5.0 | | 5.5 ± 0.7 | |
| Difference IN1 - EST1 | | | 14.2 (p=0.03) | 3.3 (p>0.5) | | |

**Table 2. Early Diagenesis Model (EDM): calibration results - model features at the two stations studied. OM1=fast-degrading POC; OM2=slow-degrading POC; OM3=mussel faeces and pseudofaeces.**

| **Station EST1** (steady-state) **(outside the influence of the farm)** | **Station IN1** (transient, 20 years run) **(under the influence of the farm)** |
|---|---|
| POC deposition: <br><br> 11.6 mmol C m$^{-2}$ d$^{-1}$ (OM1+OM2) **calibrated** | POC deposition: <br><br> 11.6 mmol C m$^{-2}$ d$^{-1}$ (OM1+OM2) <br><br> **Based on calibration performed at EST1** <br><br> 8.2 mmol C m$^{-2}$ d$^{-1}$ (OM3) <br><br> **Average yearly value predicted by the deposition model at IN1** |
| 60% fast-degrading (10 y$^{-1}$); 40% slow degrading (0.01 y$^{-1}$) <br><br> $$\frac{OM_1}{(OM_1 + OM_2)} = 0.6$$ | 38% faeces (20 y$^{-1}$); 37% fast-degrading (10 y$^{-1}$); 25% slow degrading (0.01 y$^{-1}$) <br><br> $$\frac{OM_1}{(OM_1 + OM_2)} = 0.6$$ |
| Bioirrigation rate (see profile in Fig. A3 - black color): <br><br> $\alpha_0 = 20$ y$^{-1}$ (bioirrigation rate at the interface) **calibrated** <br><br> $x_{irr1} = 20$ cm (depth of the bioirr. layer) **calibrated** <br><br> $\lambda_1 = 0.1$ cm (shape coefficient) fixed a-priori <br><br> $$\alpha(z) = \alpha_0 \cdot \frac{e^{\left(\frac{x_{irr1}-z}{\lambda_1}\right)}}{\left[1 + e^{\left(\frac{x_{irr1}-z}{\lambda_1}\right)}\right]}$$ | Bioirrigation rate (see profile in Fig. A3 - gray color): <br><br> $\alpha_0 = 40$ y$^{-1}$ (bioirrigation rate at the interface) **calibrated** <br><br> $x_{irr1} = 15$ cm (depth of the bioirr. layer) **calibrated** <br><br> $\lambda_1 = 0.1$ cm (shape coefficient) fixed a-priori <br><br> $$\alpha(z) = \alpha_0 \cdot \frac{e^{\left(\frac{x_{irr1}-z}{\lambda_1}\right)}}{\left[1 + e^{\left(\frac{x_{irr1}-z}{\lambda_1}\right)}\right]}$$ |
| Bioturbation rate: <br><br> $Db_0$=1.0 cm$^2$ year$^{-1}$; $\lambda_2 = 1.5$ cm (Mugnai et al., 2003) <br><br> $Db(z) = Db_0 \cdot e^{(-z/\lambda_2)}$ | Bioturbation rate: <br><br> $Db_0$=1.0 cm$^2$ year$^{-1}$; $\lambda_2 = 1.5$ cm (Mugnai et al., 2003) <br><br> $Db(z) = Db_0 \cdot e^{(-z/\lambda_2)}$ |
| C:N:P 129:18:1 (Brigolin et al., 2009) | C:N:P 129:18:1 (Brigolin et al., 2009) |
| Porosity parameters $\varphi(z)= \varphi_\infty+( \varphi_0 - \varphi_\infty)\ e^{-\tau z}$ <br><br> $\varphi_0$=0.77; $\varphi_\infty$=0.51; $\tau$=0.65 | Porosity parameters $\varphi(z)= \varphi_\infty+( \varphi_0 - \varphi_\infty)\ e^{-\tau z}$ <br><br> $\varphi_0$=0.82; $\varphi_\infty$=0.53; $\tau$=0.1 |
| Physico-chemical parameters and boundary conditions <br><br> T = 10°C (yearly average) <br><br> S = 36.3 PSU (measurements st. EST$_1$) <br><br> [O$_2$] = 223.5 µM (measurements st. EST$_1$) <br><br> [NH$_4^+$] = 1.0 µM (measurements st. EST$_1$) <br><br> [NO$_3^-$] = 2.5 µM (Solidoro et al., 2009) <br><br> [SO$_4^{2-}$] = 28.7 mM (measurements st. EST$_1$) | Physico-chemical parameters and boundary conditions <br><br> T = 10°C (yearly average) <br><br> S = 36.3 PSU (measurements st. IN$_1$) <br><br> [O$_2$] = 229.8 µM (measurements st. IN$_1$) <br><br> [NH$_4^+$] = 9.85 µM (measurements st. IN$_1$) <br><br> [NO$_3^-$] = 2.5 µM (Solidoro et al., 2009) <br><br> [SO$_4^{2-}$] = 28.8 mM (measurements st. IN$_1$) |
| Mineralization (Total Organic Carbon): | Mineralization (Total Organic Carbon): |

| | |
|---|---|
| Total: 11.1 mmol C m$^{-2}$ d$^{-1}$ | 18.7 mmol C m$^{-2}$ d$^{-1}$ |
| Degraded by O$_2$: 27% | Degraded by O$_2$: 30% |
| Degraded by NO$_3^-$: 1% | Degraded by NO$_3^-$: 2% |
| Degraded by SO$_4^{2-}$: 71% | Degraded by SO$_4^{2-}$: 68% |

5    **Figures**

Total: 11.1 mmol C m$^{-2}$ d$^{-1}$

Degraded by O$_2$: 27%

Degraded by NO$_3^-$: 1%

Degraded by SO$_4^{2-}$: 71%

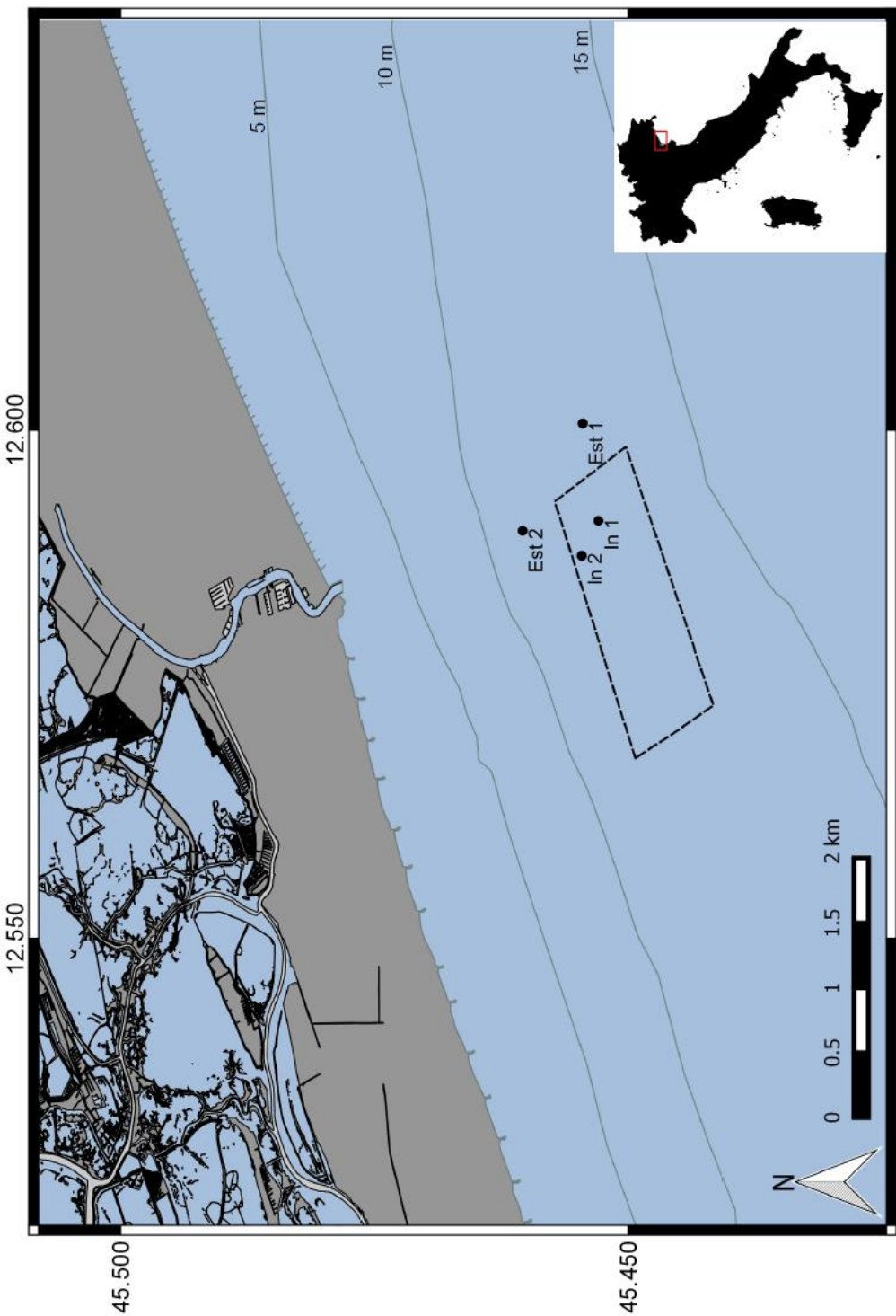

**Figure 1**

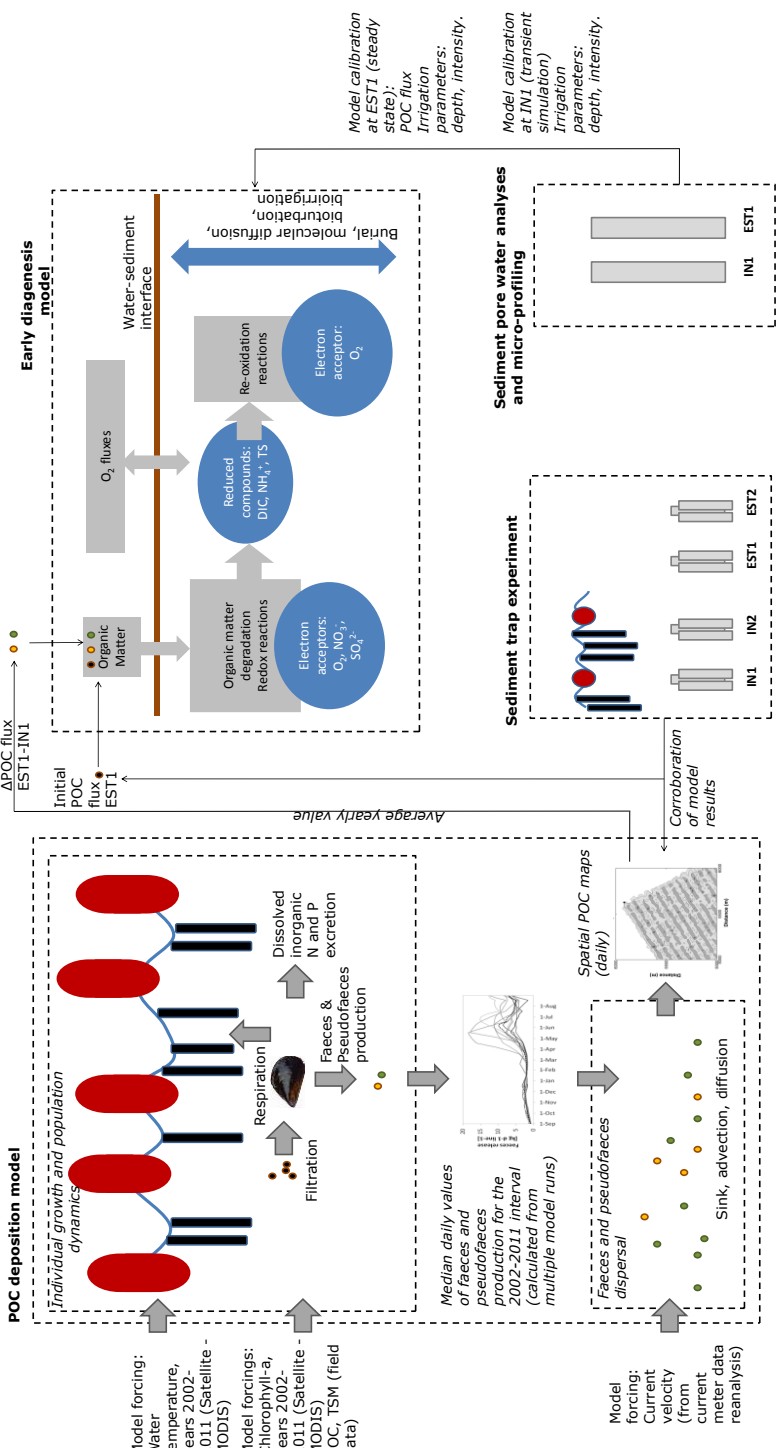

**Figure 2**

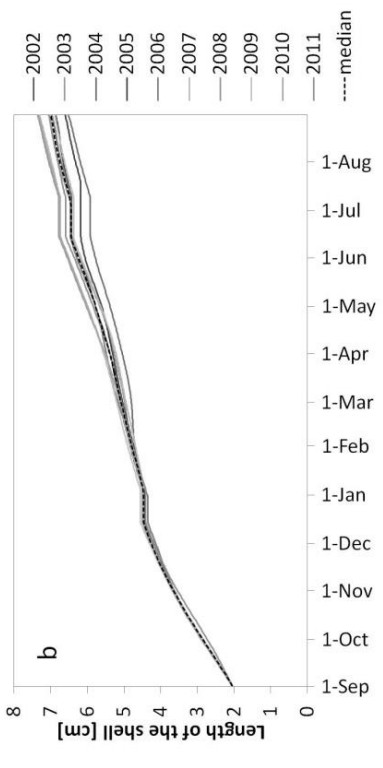

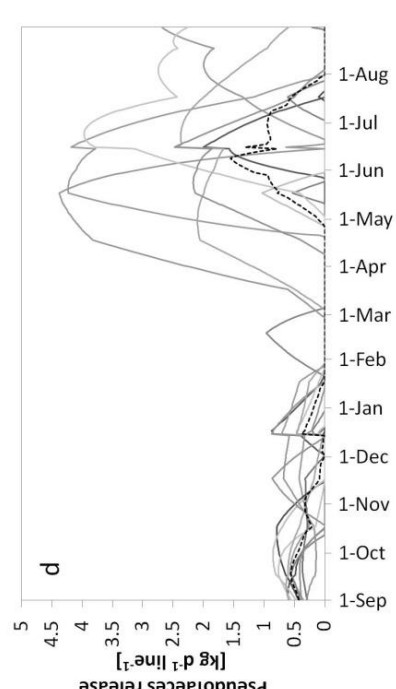

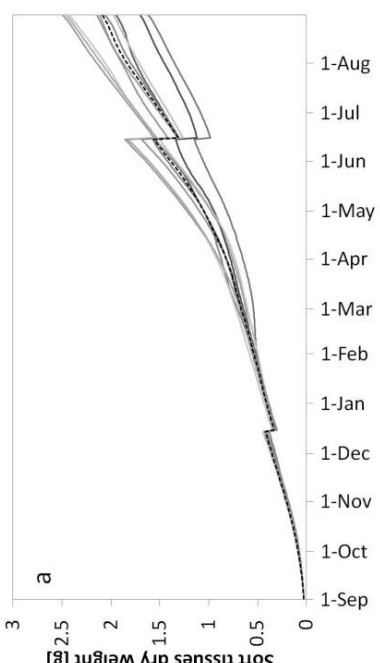

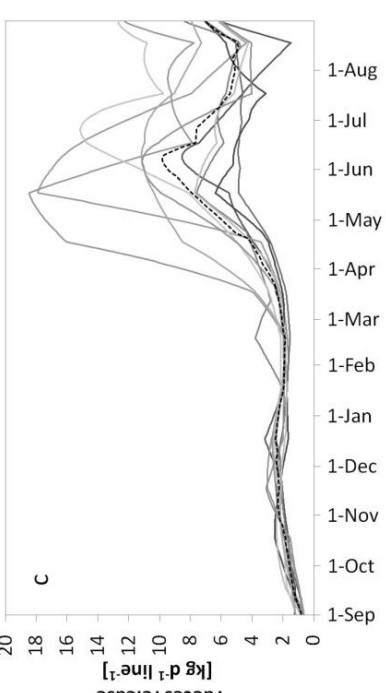

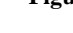

**Figure 3**

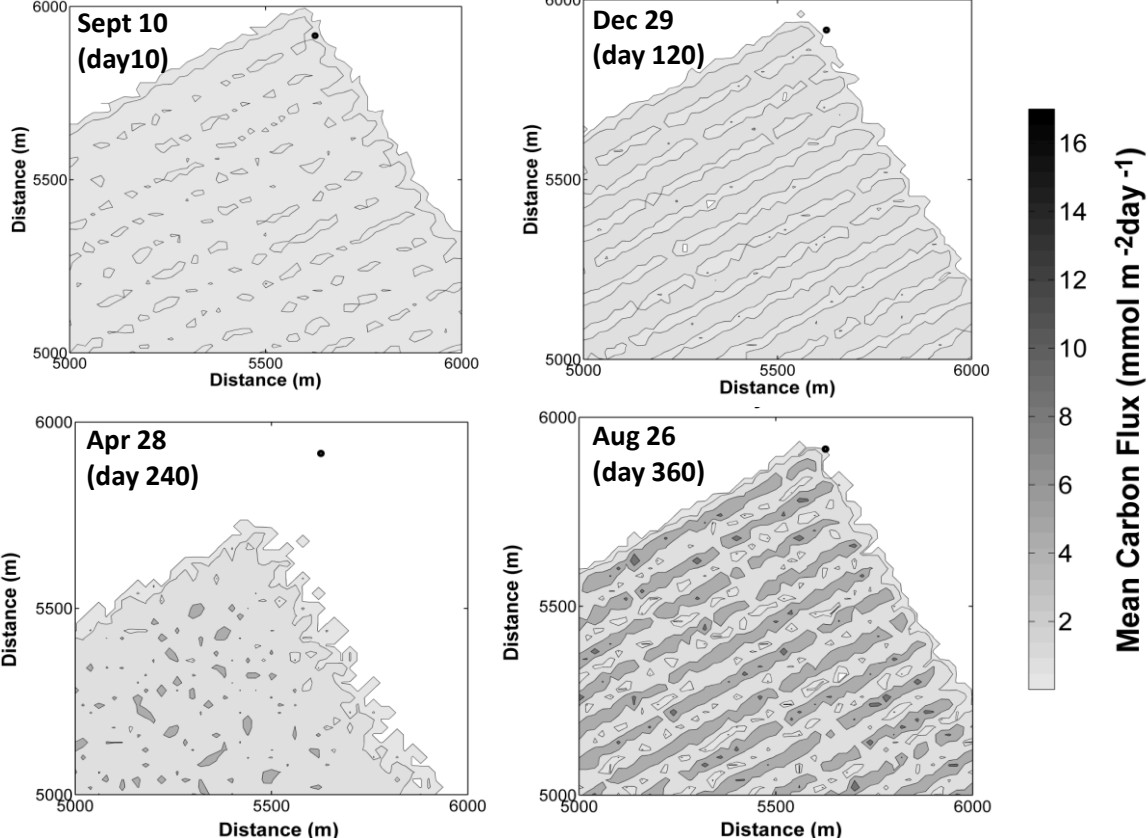

**Figure 4**

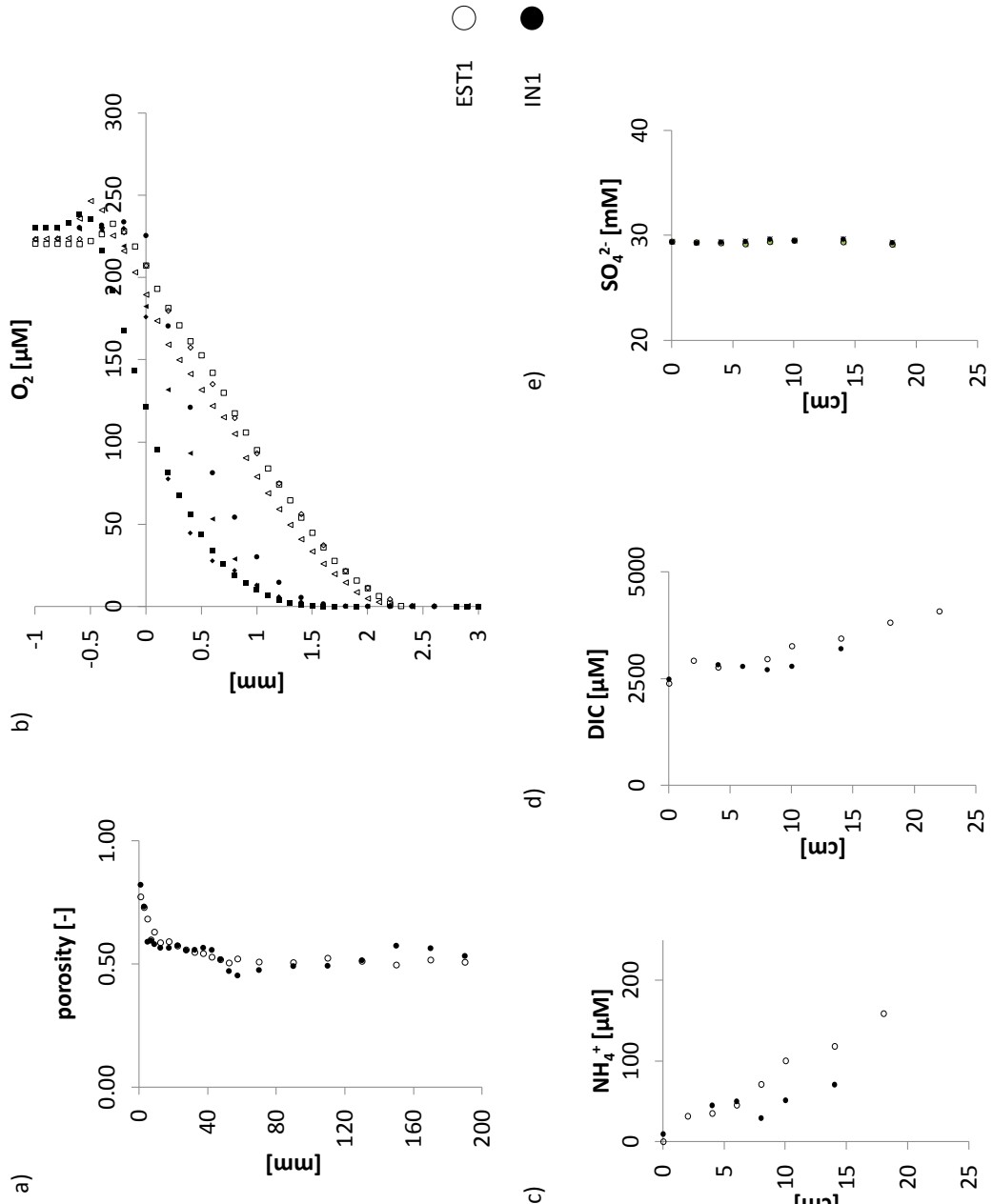

**Figure 5**

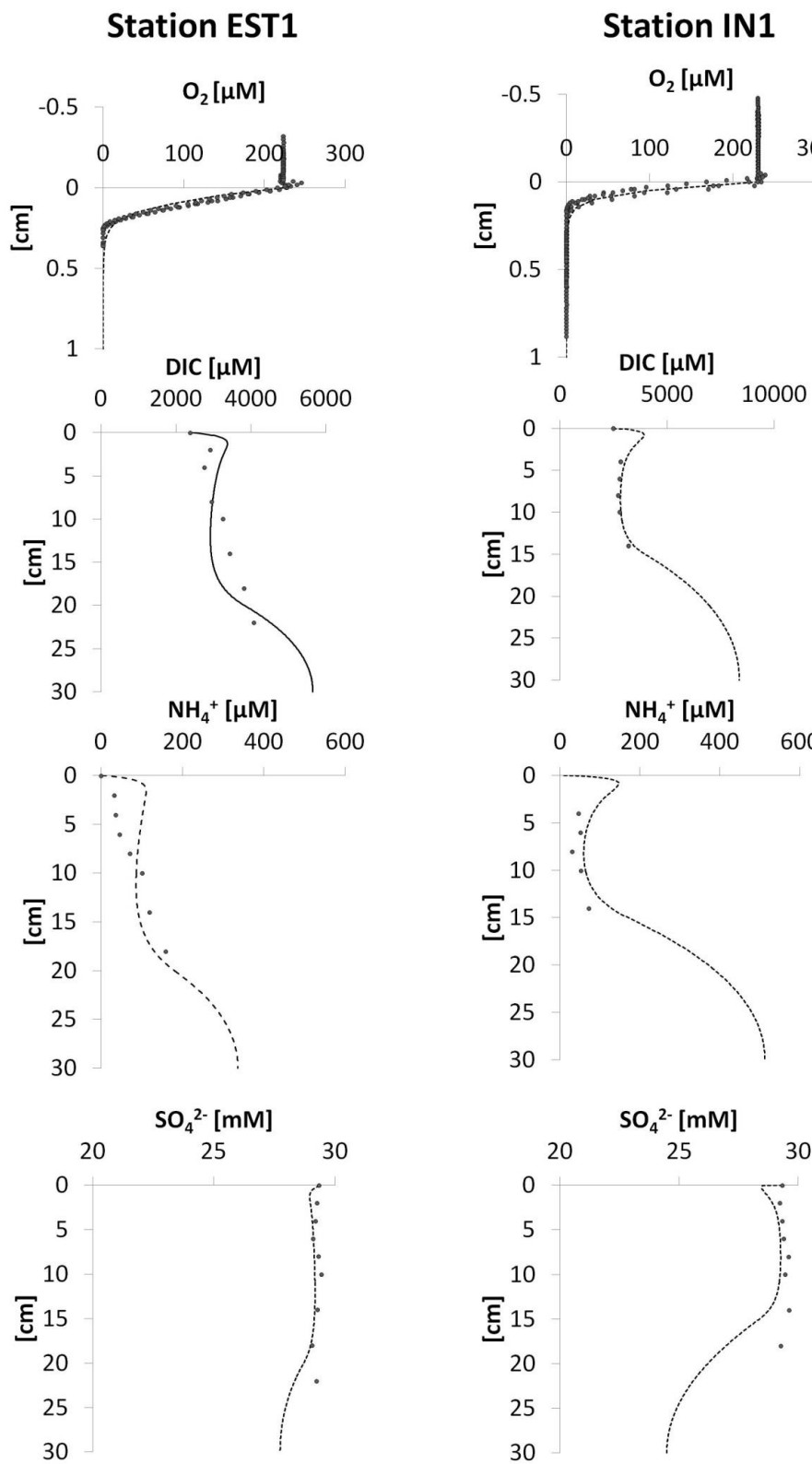

**Figure 6**

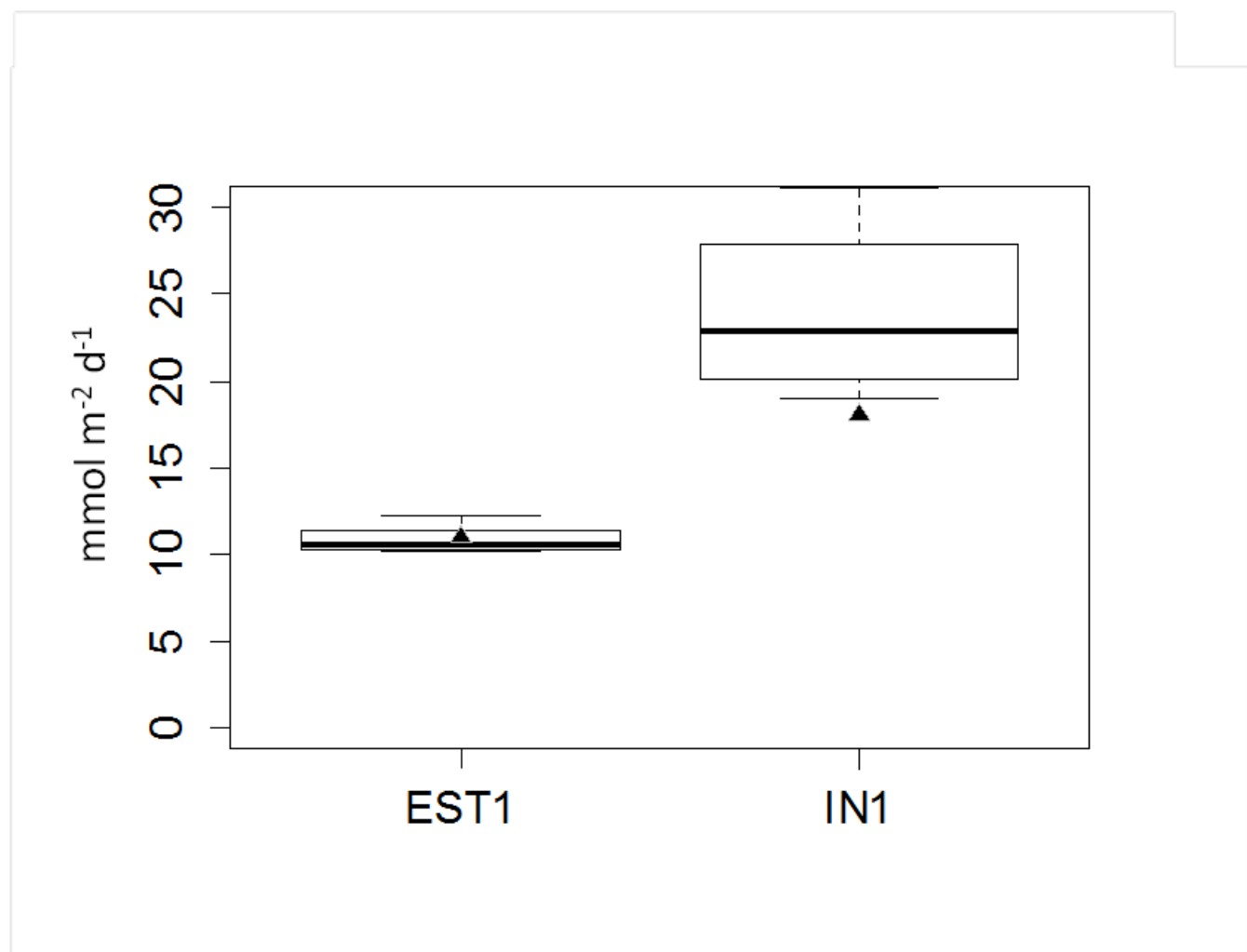

**Figure 7**

**Figure 8**

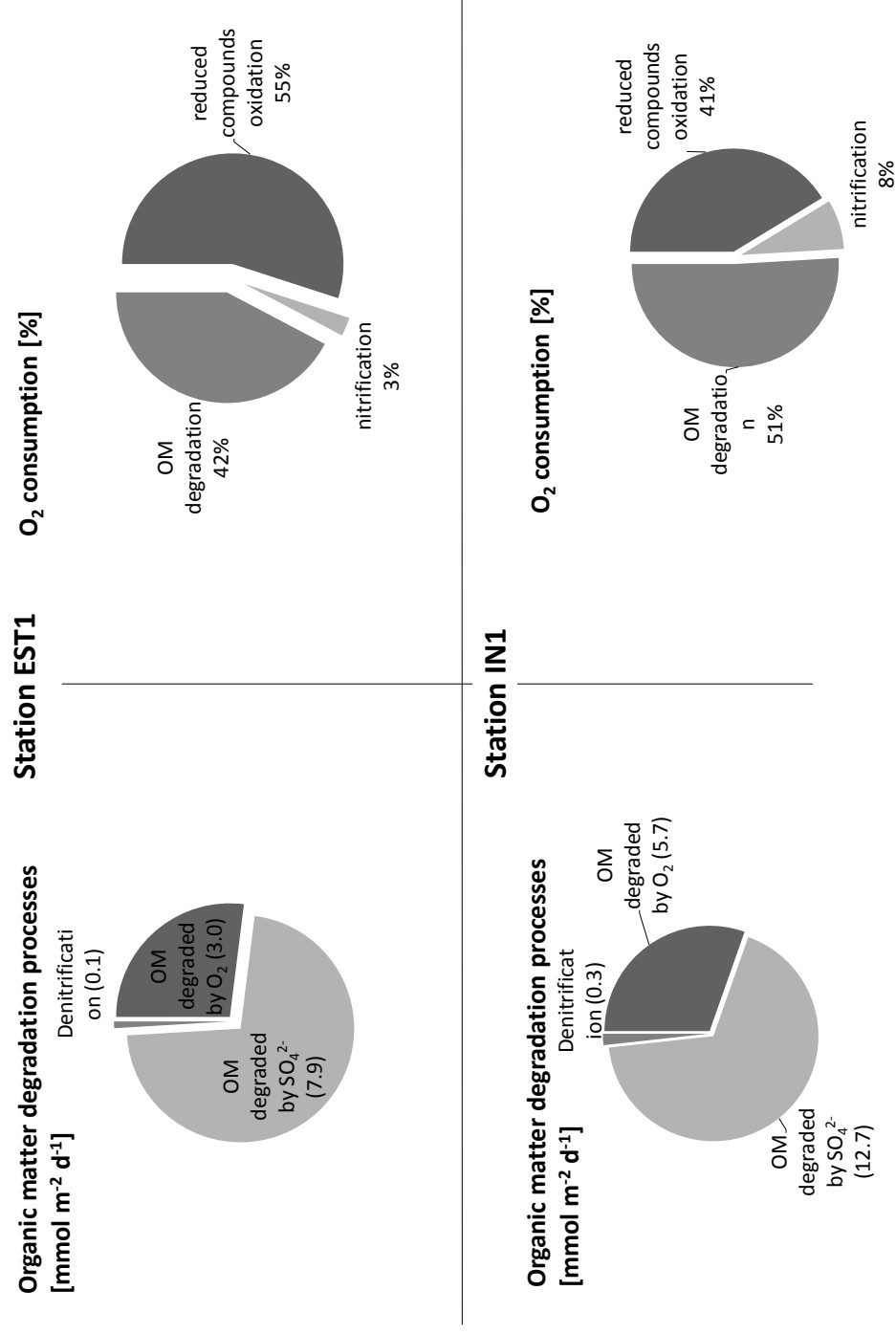

