# Peer review of "Modelling biogeochemical processes in sediments from the north western Adriatic Sea: response to enhanced particulate organic carbon fluxes"

_Biogeosciences, 2017_

## Referee Comment (RC1) · Anonymous Referee #1 · 4 Jul 2017

The present version of the manuscript is relatively well written and the objectives are well stated. The subject is very interesting and the approach used may be appropriate. However, the methods are not clearly detailed enough to allow the reader to appropriately follow what is done and how. In addition, there are many points on the form that are confusing: the use of different parameters to define (potentially) the same thing that may have different meaning in the literature (e.g., POC, Corg, OC%, . . .). The caption of the figures/tables that are not in agreement with what is written in the text or what is seen on the figure/table, . . . I tried to address most of them in the specific comment section below.

[Figure]

Overall, my most important comments concern: 1) the way bioirrigation is accounted in the early diagenetic model, which is one of the two parameters used to calibrate the model 2) the representativity of one porewater profile (and of 1 or 2 porewaters samples) to entirely support the bioirrigation process 3) the representativity of the measured O2 profiles and derived diffusive flux carried out under ex situ conditions without any stated precautions to be representative of in situ conditions. All the discussion/conclusion relies on those simplifications and potentially non representative data, that strongly limits its credibility.

Specific comments: Title: - define POC in the title or if you assume that it is 100% evident, it should be at least defined in the abstract and in the manuscript the first time it is used? The only place where it is defined is at L27, p3, clearly too late for the reader.

Introduction - L2, p2: please, precise the difference between faeces and pseudo-faeces? Do you assume that both materials are included in OM3 pool (deposited pool) defined later? It may help the reader to better understand. - L6, p2: replace the last coma by "and" - L9, p2: correct "Estuarine" - L14, p2: what "degree of deposition" means? Is it a characteristic of the deposition which is influenced by the local hydrodynamic?

Materials and Methods - L29-39, p2: This part is a little bit confusing: it seems to announce the organization of the Materials and Methods section but the following parts do not follow such structuration. There is no information allowing to link the POC production and the POC deposition (i.e., the deposition model). I suggest to improve this part in order to fit the following parts of the section (potentially separating the modeling approaches and the experimental approaches used to calibrate the models), or to remove it from here and resume it at the end of the introduction section. - L3, p3: remove "see" before Figure (to be applied to all the manuscript) - L18-19, p4: What is the impact of neglecting Fe and Mn biogeochemical related processes on the total biogeochemical processes? Are these processes really negligible? This could be the

case as sulfate reduction process appear negligeable (from $SO_4^{2-}$ profiles and in spite what your modeling result tend to indicate). In addition you do not have any $NO_3^-$ data that could also be an important process in the area studied. - L1, p5: what is POM? Precise the relation we POC you assume. - L1, p5: what is TSM? Is it the same as TSS? Please homogenize throughout the manuscript and if there is a difference between these parameters, clearly explain it. - L2, p5: Why do you cite Rampazzo et al., 2013 as you mention that POC/TSM was extracted from the same previously cited paper (i.e., Brigolin et al., 2009)? - L2, p5: what is AE? - L5-6, p5: what is the influence of the farm on the current within the farm? This is the current within the farm that will drive the transport and deposition of particulate from the farm. This is not explained within the deposition model section. I think this is important for the reader. Please explain. - L5-10, p5: How the annual variability of the current was obtained as only 6-7 months (March-September) of measurement is available? - L15, p5: I guess you refer to the POC downward flux here ("Initial values of POC for the calibration...")? Please precise. - L17-18, p5: what do you mean by transient conditions? Is it the second EDM model that is mentioned at L4, p2? Is it a completely different one? - L21, p5: Diffusive $O_2$ fluxes were assessed from the profile in the surface sediment? Accounting for water temperature and salinity, and tortuosity in sediment? Please precise - L24-25, p5: it appears that the end of the rearing cycle is the last days of August while the beginning is early September. Does this mean that the harvesting of the mussels occurs within these few days each year? Does the activity of harvesting induce a strong increase of the concentration of suspended matter that may remain several days within the water column and may impact the material trapped within the sediment traps deployed at the beginning of the new rearing cycle? - L34-35, p5: only one sediment core was collected per station. How the spatial heterogeneity can be addressed with only one core? - L35-36, p5: Did the profiling experiment carried out at in situ temperature? If not, please precise at least the in situ vs ex situ temperature. - L35-36, p5: Does the bubbling was performed using ambient air? Is it coherent with the in situ oxygen concentration? - L41, p5: how the porewaters were extracted?

Results L17, p6: What is CV? L21, p6: why to chose days 10 and 360? Because they correspond to the min and max situation? Precise. L22, p6: why the footprint is related to the presence of lines here? The shape of the farm appear also clearly on the deposition map (with or without the lines visible). L22, p6: "is clearly visible at days 120 and 360" It is also the case the day 10. L25, p6: OC% not defined and no information about its analysis. In Table 1 "TSS" seems to be Total mass fluxes, OC% seems to be Corg(%) and POC flux seems to be Total Carbon flux? Please be consistent all throughout the manuscript. L30, p6: if you decide to call the station outside the farm a "reference station" please use this terminology all through the manuscript. L33, p6: no information about how porosity was measured. L36, p6: "Oxygen shows a quasi-monotonous decrease in concentration" you mean downward in the sediment? L37, p6: what do you mean by low variability? L4-5, p7: I clearly doubt that the trend in the only one DIC profile (and the two subsurface samples) reported can indicate anything about bioirrigation. L7, p7: O2 with 2 in index (same all through the manuscript) L13-14, p7: Is the bioirrigation expected to decrease the porewater concentrations at a precise depth? I think there is a need to better explain how bioirrigation is taking into account in the model. Usually, the bioirrigation rate is applied over a depth interval with the intensity decreasing (linearly or not depending the model) with depth from a maximum value at the surface to a 0 value at a specific depth. This induces a dilution effect of the porewater by the overlying water decreasing with depth. Is this not the case here? If not, please precise this aspect. In addition, information on the macrofauna species and bioturbation behavior may help to define the way to account for bioturbation (sediment reworking + bioirrigation) processes (in Colla, 2017?). L14, p7: I'm not agree with this affirmation. The range of concentration between measured and modeled profiles are similar but the vertical trends clearly differ. L17-20, p7: Figure 7 seems to show the diffusive O2 fluxes for IN and EST stations. I can't see any comparison between modeled and measured data. L17-20, p7: how the measured ex situ O2 profiles, and assessed O2 diffusive fluxes, may be used here to gain information on the oxygen uptake in situ since: 1) profiles seems to haven't been performed at in situ temperature

and in situ oxygen content, and 2) under sunlight influence that could be clearly higher than under in situ condition (observation microphytobenthos production that is know to strongly impact the diffusive fluxes at the sediment-water interface as well as the O2 penetration depth). This is a crucial step that may to be clearly addressed. - L22-23, p7: the three pools of OM: you actually mean the total organic matter that correspond to 2 pools in EST station and 3 pools in IN station? - L24-25: this is not what is reported in Figure 7.

Discussion L4, p8: What is ED? It should be EDM as previously L24-32, p8: You said that modeled POC flux from EDM agrees with POC flux measurements from a sediment trap. (that already was the subject of the previous section) but then try to explain why there is a factor 2 difference. L12-13, p9: This is an important point.

Table 1: - title do not correspond to what can be seen in the Table - please add a mention specifying the stations that are inside the influence of the farm and those outside the farm (reference?) and gather results for stations IN and EST to help the reader. - be careful to the significant digits. - homogenize the position of the text inside the cell

Table 2: - Second line corresponds to POC deposition I guess. This should be clearly mentioned. - please add a mention specifying the data for inside the influence of the farm and this outside the farm (reference?) to help the reader. - How can you assume that the CNP ratios are the same in both stations. - mineralization rates reported correspond to total (OM1 + OM2 + OM3) organic carbon?

Figure 2: The caption is clearly no detailed enough? What are the red bubbles? The black rectangles? The blue lines?

Figure 4: Please add the simulation day on the figure. It will help the reader to follow the writing.

Figure 5: - specifiy "por" in the caption or write "porosity" on the figure. Same for DIC.

- I doubt the unity of porosity is % as mentioned.  - the term "micro-profile" is only applicable to O2 and eventually to porosity profiles.

Figure 6: precise this caption.

Figure 7: - caption of the figure 7 is unrelated to the figure 7 - write on the figure (noted Figure 7) that it correspond to O2 diffusive fluxes (if this is well the case).

[Figure]

---

## Referee Comment (RC2) · Anonymous Referee #2 · 10 Jul 2017

General Comments

First let me apologies for the late submission of these comments as I lose the original document with the formal comments. This is a quick summary of what I remember by going through the draft again. Additionally let me add to this that my experience is with physical marine numerical modelling and data analysis, and I am not particularly familiar with the application on mussel farming. These aspect will be noticed by the focus of my comments.

The approached ideas and objectives are interesting, however I see issues (disagree) with some of the method and ways that the data was used. This could be as I am

unfamiliar to this particular application and a good answer could quickly clarify these aspects (see specific comments). At this point I will not recommend to publish this work if not further improvements are completed.

It was difficult to follow the document at the beginning (until section 2.3), after this it was easier to follow the structure. However improving figures 1 and 2 will help to create a clear big picture at the start (please, see specific comments).

The specific comments and technical corrections start with the page (P) number and line (L) number to indicate the specific place in the text that is commented.

Specific Comments

P4 L38: How reliable is the Chla and SST data when the study area is near the coast (at ∼2km and the data resolution is 4km)? If you imply that the average of certain number of Chl-a and SST data points is representative of the study area, Where is explain in the methodology? For the coordinates mentioned in P4 L37 the number can be guessed, but it will be better to see the number. And finally, Is it representative for the 2km2 study area the Chla and SST time series if they were constructed based on an area of app. 100Km2?

P4 L38: How do you address the problem of color particulate matter, optical shallow waters and water turbidity, where the river could have a major role to play? As example please see: * Cannizzaro and Carder (2006)† [optical shallow waters]

P5 L6: Could you show the current meter data?, as you say that you are using the residual current edited. Why did you add storms? How many storms did you add in this random process? (You say that this method should be preferred for forecasting, however you are reproducing a period where you have data to validate and calibrate, Is not this closer to a hind-casting than a forecasting)

Why you are not using the current due to tides? Tidal currents will have an impact on particles that represent the faeces and pseudofaeces

P6 L11: Please explain further in the method section that the mean of Chl-a and SST is spatial and temporal, resulting in a mean or characteristic year for each variable. This means that strong events of Chla are smooth as shows figure A1b, which means implication for the model individual growth and population model.

P6 L34: If you are going to use cm here, then please use cm in the figure.

P7 L36: Are the features and methods of Weise et al, (2009) comparable to yours? Depth of the study area, mean velocity (if they are considering residual current or not), size of the farm, etc The same comment goes of the Hatstein and Steven (2005) reference.

Figure 1 : The figure does not explain itself. It is not possible to know the meaning of the rectangle and and points in the figure. In my first look at the figure, the river was a road and I have to google the river to find out where it was.

Figure 2 : The figure caption is not good. Please give to the reader enough information for the "Information flow". Please add to the boxes in the diagram the reference to each represented/used model. Why are you using the word reanalysis for the current input in the particle tracking model (faeces and pseudofaeces)?

Figure 3 : What is the meaning of the sharp changes shown by graph a? It will better If the caption has more information about the figure. As example, if this represented data are modeled results or measurements.

Figure 5: Squares and triangles without legend. Y-axis without label just units.

Figures in general: Improve caption for the figures to stand by themselves.

P6 L23: Does this agree with the mean current velocity, mean vertical velocity for the particles to sink and mean depth?

Technical Corrections P3 L27: Define acronym POC earlier. There are few times before this where the acronym is used.

P4 L16: Reference Cappellen and Wang, 1996 is not listed in the references

P5 L 26: Define the acronym PVC.

P6 L6: Define acronym HPLC

Boudreau (1996) and Sanchez-Jerez et al (2016) listed in the reference but I did not find them in the text

---

## Author Comment (AC1) · 27 Jul 2017

The present version of the manuscript is relatively well written and the objectives are well stated. The subject is very interesting and the approach used may be appropriate. However, the methods are not clearly detailed enough to allow the reader to appropriately follow what is done and how. In addition, there are many points on the form that are confusing: the use of different parameters to define (potentially) the same thing that may have different meaning in the literature (e.g., POC, Corg, OC%,...). The caption of the figures/tables that are not in agreement with what is written in the text or what

is seen on the figure/table,... I tried to address most of them in the specific comment section below.

We thank the reviewer for all the detailed and useful comments made. Our answer to the three general points, and to specific points related are reported below. We tried to track within this document most of the changes performed on the text, when this was not possible, changes were implemented only on the manuscript (ms), and reference reported in the present document. Reference to the initial submission and to the updated version of the manuscript were detailed. When not stated, we imply reference to the revised version. The revised version of the ms is provided as a supplement.

Overall, my most important comments concern: 1) the way bioirrigation is accounted in the early diagenetic model, which is one of the two parameters used to calibrate the model With respect to this point, in the new updated version of the manuscript (ms), the following elements were included: - The formulation used to define the profile of the bioirrigation rate has now been added in Table 2 (formulation for bioturbation was also specified in Table 2); - A sentence pointing to the methodology used to include bioirrigation in the BRNS tool, used for developing the EDM was added in the methods section (ln 7-11, pg 4); - A plot showing the shape of bioirrigation rates in the two profiles was included in the appendix (Appendix, Figure A3)

2) the representativity of one porewater profile (and of 1 or 2 porewaters samples) to entirely support the bioirrigation process

We acknowledge that 2 cores cannot define very strongly bio-irrigation in this area, but, as this process is everywhere in coastal sediments, and we report numerous macrofauna, bio-irrigation should be active there. In order to support this point, as suggested by the reviewer, we included in the text a list of species which were found at the two stations within the thesis work by Colla (2017), and that we expect to play an active role in bioirrigation. Bioirrigators were ranked based on the reworking potential of each taxa, according to the approach proposed by Solan et al. (2004).

We would like to remark that two sets of profiles indicating the same type of distribution were available for each core: DIC and NH4, moreover the effects could be corroborated by the shape of the profile of another chemical species (SO42-), available from the same cores. This was stated in the ms (page 9 line 15, previous ms version):"The shape of the DIC–NH4+ profiles indicates bio-irrigation (Meile et al, 2001; Canavan et al., 2006), although the deeper increase is not visible in our data profiles due to limited penetration of the cores. Indeed, the very limited increase in concentration profiles in the first centimeters can only be linked to input of bottom water with lower DIC and NH4+ by irrigation, given the large recycling intensity in surface sediments as exemplified by O2 profile."

Based on the reviewer comment, we also edited the conclusions, in order to communicate that uncertainty on the coefficients could be large, due to the lack of data. The sentence is the following: "Indeed, the early diagenesis model calibration led to an estimation of enhanced and more shallow bioirrigation underneath the farm which were confirmed by independent data on macrofauna composition collected at the study site. We remark that, based on the number of cores available, it was not possible to assess quantitatively the uncertainty related to these coefficients, which estimation would allow to characterize more strongly bioirrigation in this area."

Solan, M., Cardinale, B. J., Downing, A.L., Engelhardt, K.A.M., Ruesink, J.L., Srivastava, D.S., 2004. Extinction and Ecosystem Function in the Marine Benthos. Science 306, 1177-80.

3) the representativity of the measured O2 profiles and derived diffusive flux carried out under ex situ conditions without any stated precautions to be representative of in situ conditions. All the discussion/conclusion relies on those simplifications and potentially non representative data, that strongly limits its credibility.

Oxygen microprofiles were measured with many precautions. First, we made sure that the cores recovered for the microprofile analysis were undisturbed, they were kept

closed until the turn in the harbor where the field camp was organized. As the water temperature and the air temperature did not vary by more than 2°C at this time of the year with the cloudy conditions present in these sampling days, we did not use the cryostat to control temperature but monitored the temperature at the start of the experiment and at the end. Furthermore, as these waters are near saturation with oxygen, we bubbled air (as stated in "Material and Methods" section) in the core in order to gently stir and to maintain a constant O2 concentration at saturation in the overlying water. This procedure was already used by our grouop with success in previous publications (see Khalil et al., 2013, Aquatic Geochemistry; Cathalot et al., 2015, PlosOne) We rewrote the Material and Method section in order to introduce these elements: "Sediment were sampled at stations IN1 and EST1 in June 2015 (respectively on 23/06 and 24/06). Undisturbed cores were collected by means of an Uwitec corer (10 cm diameter; 20 cm avg. penetration depth). Water was sampled 2 m above the bottom by means of a Niskin bottle, for dissolved oxygen, salinity and temperature determinations. Cores were immediately brought back to the field camp and prepared for microprofiling, which was conducted a few hours after coring. As the temperature of the outside air was within a few degrees of the water teperature during the cloudy sapmpling days (23°C in air versus 21°C in the water), the temperature was not controlled using the available cryostat, but monitored at the start and end of the measurements, and showed minimal variations. Cores were bubbled with air during measurements to allow aeration and gentle stirring. As the bottom waters were saturated with oxygen, bubbling maintained the proper in situ O2 conditions. Microprofiling was conducted with a Unisense motorized microprofiler. Four oxygen microprofiles were performed using 100 $\mu$m tip microsensors which were calibrated by a two-points method: Winkler titration of the overlying water (with a precision of 2 permil) and zero-oxygen signal in the anoxic layer below the oxic zone."

Specific comments: Title: - define POC in the title or if you assume that it is 100% evident, it should be at least defined in the abstract and in the manuscript the first time it is used? The only place where it is defined is at L27, p3, clearly too late for the

reader. Thanks. Particulate Organic Carbon was reported in the title. POC acronym was defined within the abstract, as well as in in the introduction.

Introduction L2, p2: please, precise the difference between faeces and pseudo- faeces? Do you assume that both materials are included in OM3 pool (deposited pool) defined later? It may help the reader to better understand. –

The sentence was modified as follows: "However, the production of faeces and pseudofaeces (excess particles rejected by palps before ingestion) leads to a net transfer of organic matter from the water column to the surface sediment"; correct, both materials are included in the same OM3 pool, this was better specified at line 14 pg 4.

L6, p2: replace the last coma by "and" corrected

L9, p2: correct "Estuarine" corrected

L14, p2: what "degree of deposition" means? Is it a characteristic of the deposition which is influenced by the local hydrodynamic? Yes, in order to be more precise the sentence was rephrased, and now reads as follows: "Based on these works it was possible to have a clearer mechanistic understanding of the relationship between the values of flux and the area affected by organic deposition and the different farming conditions (in terms of local hydrodynamics and farm characteristics – depth; geometry)."

Materials and Methods

L29-39, p2: This part is a little bit confusing: it seems to announce the organization of the Materials and Methods section but the following parts do not follow such structuration. There is no information allowing to link the POC production and the POC deposition (i.e., the deposition model). I suggest to improve this part in order to fit the following parts of the section (potentially separating the modeling approaches and the experimental approaches used to calibrate the models), or to remove it from here and resume it at the end of the introduction section. – Thanks for this useful suggestion. Lns 29-39 were integrated in the last part of the introduction, by extending the content of lines 21-24, pg 2 (original lines numeration). The text now reads as follows: "In this work, a longline mussel farm located in the north western Adriatic Sea was regarded as a local source of perturbation of natural organic matter downward fluxes. Average yearly increase in Particulate Organic Carbon (POC) flux induced by the mussel farm throughout the year was first quantified by applying a biogeochemical model of POC production and deposition (mussel faeces and pseudofaeces), coupled to two sediment trap deployments, which were carried out at the beginning and at the end of mussels farming cycles, with the aim of corroborating model predictions. Outputs of this first model, were subsequently used in early-diagenesis model simulations (one steady-state and one transient), which were constrained by the observed field data in the sampled cores at two stations (pristine and impacted): bioirrigation parameters and ratio among degradation pathways were estimated on the basis of model application. Measurements included O2 micro-profiling, porosity and micro-porosity, pore waters NH4+, SO42-, and Dissolved Inorganic Carbon (DIC)."

L3, p3: remove "see" before Figure (to be applied to all the manuscript) –

change implemented

L18-19, p4: What is the impact of neglecting Fe and Mn biogeochemical related processes on the total biogeochemical processes? Are these processes really negligible? This could be the case as sulfate reduction process appear negligeable (from SO42- profiles and in spite what your modeling result tend to indicate). In addition you do not have any NO3- data that could also be an important process in the area studied. -

Thanks. Including Mn and Fe related processes would definitely increase the EDM realism, but also its complexity. Mn and Fe dynamics were present in a previous modelling work applied in a close area, located in the Northern Adriatic (Brigolin et al., 2011, cited in the text). However, with respect to the present work, in this other case, a larger database was available for constraining the model (pg 4, lns 18-20 - initial version of the ms). This model limitation was stated explicitly in the methods section of the ms,

and a remark concerning the potential development of the model was also present in the discussion (pg 9, lns 43-44 - initial version of the ms) " A more precise estimation of the fate of this oxygen could be obtained by introducing in the model FeS precipitation, for which at least Fe2+ measurements in pore waters would be required.". With respect to NO3, we acknowledge that this would be a relevant information for constraining nitrification/denitrification, and this was highlighted in the discussion (pg 10, lns 5-9 - initial version of the ms) " Higher NH4+ concentration predicted at station EST1 with respect to field data could be explained with a higher rate of nitrification at this station. However, in the calibration performed within this work, the kinetic constant for nitrification was kept at its original value (Table A4), due to the lack of data concerning NO3-."

L1, p5: what is POM? Precise the relation we with? POC you assume. – We added POM and TSS extended name. TSM was replaced by TSS, in order to homogenize this term throughout the manuscript (see our reply below). POM was determined independently, as part of the survey described in Brigolin et al. (2009), and not by assuming a defined POC:POM ratio.

L1, p5: what is TSM? Is it the same as TSS? Please homogenize throughout the manuscript and if there is a difference between these parameters, clearly explain it. Correct. This was checked, see the question above.

L2, p5: Why do you cite Rampazzo et al., 2013 as you mention that POC/TSM was extracted from the same previously cited paper (i.e., Brigolin et al., 2009)? Thanks. For the sake of clarity the citation was removed.

L2, p5: what is AE? Absorption Efficiency (AE)

L5-6, p5: what is the influence of the farm on the current within the farm? This is the current within the farm that will drive the transport and deposition of particulate from the farm. This is not explained within the deposition model section. I think this is important for the reader. Please explain. This engineering aspect, although of interest, has not been properly investigated yet. To the best of our knowledge, multiple current meter

measurements within the same longline suspended mussel farm are not available, and model predictions regarding the local effect of structures on currents would be complex to validate. Similar effects are currently neglected in most deposition models also for fish cages (which nets are expected to have more relevant impacts on the current flows). We added a synthetic remark on the fact that the aspect is not included in the model (model theory paragraph 2.2, ln 35 pg 3).

L5-10, p5: How the annual variability of the current was obtained as only 6-7 months (March-September) of measurement is available? – Thanks. In order to perform this step, we followed the outline adopted by Jusup et al (2007). In the new version of the manuscript additional details were included in the description of the methodology, based on the suggestions by reviewer #2. The text now reads as follows (p5): " Modelling deposition requires an input time series of water velocity at an hourly time step. These data were provided on the basis of a current meter deployment carried out between March and September 2010 at a station located approximately 500 m from the NE edge of the farm (Boldrin A. pers. comm., see Fig. A2). Current meter data were first processed by means of a classical harmonic analysis, in order to extract tidal components as well as long-term residual means (Pawlowicz et al., 2002). On the basis of the procedure proposed by Jusup et al. (2007), the residual currents were therefore edited randomly for short periods of time in order to reproduce the variability recorded from current meter measurements during extreme events (i.e. storms). Number of events was imposed on the basis of the 2010 current time series, and of previous current meter deployments available for this area (Rampazzo et al., 2013; Giovanardi et al., 2003). Effects of tide and storm events were therefore accounted in the final time series, while short-period fluctuations related to turbulence were accounted for by the deposition model, as reported by Jusup et al. (2007)."

L15, p5: I guess you refer to the POC downward flux here ("Initial values of POC for the calibration...")? Please precise. – corrected

L17-18, p5: what do you mean by transient conditions? Is it the second EDM model

that is mentioned at L4, p2? Is it a completely different one? Correct. The same model structure was used, although boundary conditions, and initial conditions, were specific for this station, and irrigation parameters were independently calibrated for this site. We added a reference to this in the last part of the introduction (Ln 28 pg 2), and within this section (Ln 19 pg 5) "The model, which had the same structure of the EDM run at EST1, was run for 20 years (time of activity of the farm) . . .".

L21, p5: Diffusive O2 fluxes were assessed from the profile in the surface sediment? Accounting for water temperature and salinity, and tortuosity in sediment? Please precise Diffusive O2 fluxes were assessed from measured oxygen profiles in the sediment by considering the oxygen gradient within the thin diffusive boundary layer. Temperature and salinity corrections were accounted for, based on measurements performed on bottom water samples. Porosity was taken into account, and the calculation of the diffusion coefficients was based on Andrews and Benett (1981).

Andrews, D. and Bennet, A.: Measurements of diffusivity near the sediment-water interface with a fine-scale resistivity probe. Geochim. Cosmochim. Acta 45, 2169-2175, 1981.

Text was changed as follows:" Diffusive oxygen uptake was calculated from profiles (both model and data, see section 2.5 below) by means of the 1-D Fick's first law of diffusion. These fluxes were assessed from oxygen profiles by considering the oxygen gradient within the thin diffusive boundary layer. Temperature and salinity corrections were accounted for, based on measurements performed on bottom water samples. Porosity was taken into account, and the calculation of the diffusion coefficients was done in accordance with Andrews and Benett (1981)".

L24-25, p5: it appears that the end of the rearing cycle is the last days of August while the beginning is early September. Does this mean that the harvesting of the mussels occurs within these few days each year? Does the activity of harvesting induce a strong increase of the concentration of suspended matter that may remain several

days within the water column and may impact the material trapped within the sediment traps deployed at the beginning of the new rearing cycle? Thanks. As reported in section 2.1, the farmed area covers about 2 km2, and mussel within this area are normally harvested within July-September, after a rearing cycle lasting a single year. This represents the average situation, which was considered for the POC production/deposition modeling, in order to conceptualize the mussel cultivation cycle and parameterize its features. In fact, the time table of activities can present some variability, according to sea conditions, market request, and farmer strategy. In the specific case, during the August 29-31, 2014 experiment mussels were still on the ropes without strong ongoing recollecting activity (we performed traps deployment over the week-end, in order to minimize the noise). The second experiment was performed one year later (September 11-13, 2015) some days after the beginning of the cycle (this second deployment was also performed over the week-end).

L34-35, p5: only one sediment core was collected per station. How the spatial heterogeneity can be addressed with only one core?

We agree with the reviewer that one core per station is not enough to capture spatial heterogeneity. It was beyond the scope of this study to understand the heterogeneity of the area and we concentrated on collecting enough porewater and solid phase data on each core to constrain the processes by using our diagenetic model. This was already a large amount of work and we could not achieve more than one core in the time of the study. We still believe that our approach is valid because of the joint use of measurements and model which allow to estimate process rates and compare them, which is a more robust and integrative output than just comparing pore water profiles.

L35-36, p5: Did the profiling experiment carried out at in situ temperature? If not, please precise at least the in situ vs ex situ temperature.

See response above, the temperature in air and in the water (23°C versus 21°C) were very close and temperature was not altered much during the measurement.

L35-36, p5: Does the bubbling was performed using ambient air? Is it coherent with the in situ oxygen concentration?

Bubbling was conducted with ambient air, as the bottom water of the sites were well-oxygenated : 223 and 229 $\mu$mol O2/l which correspond to 100% saturation in Mediterranean seawater (38 permil at 20-21°C).

L41, p5: how the porewaters were extracted?

Porewaters were extracted using Rhizons$^{®}$ (Seeberg-Elverfeldt et al. 2005) which are porous soil samplers operated with depression. The cores were sampled under N2 within 4 hours after coring. The text was modified to include this method: "Porewaters were extracted within 4 hours after coring in a glove bag under N2 using Rhizons$^{®}$ (Seeberg-Elverfeldt et al., 2005)."

Results

L17, p6: What is CV? "Coefficient of Variation", the extended name was included in the ms

L21, p6: why to chose days 10 and 360? Because they correspond to the min and max situation? Precise. Correct, we rephrased the sentence as follows: " Maximum organic carbon fluxes predicted at day 10 and 360, representative of the situation at the beginning and at the end of the growth-out cycle, are 2.5 and 13.3 mmol C m-2 d-1, respectively."

L22, p6: why the footprint is related to the presence of lines here? The shape of the farm appear also clearly on the deposition map (with or without the lines visible). With this comment we would like to point out that, based on model results, the shape of the lines is more clearly visible on the deposition footprint on some days with respect to others (which are characterized by specific conditions in term of currents, and amount of biodeposits exiting the farm - this aspect is additionally explored in the answer to reviewer #2).

L22, p6: "is clearly visible at days 120 and 360" It is also the case the day 10. With respect to this point, we are not in agreement with the reviewer, since the map at day 10, if examined alone, and not in the context of the whole figure, seems not to present a clear univocal pattern related to the presence of the mussel lines.

L25, p6: OC% not defined and no information about its analysis. In Table 1 "TSS" seems to be Total mass fluxes, OC% seems to be Corg(%) and POC flux seems to be Total Carbon flux? Please be consistent all throughout the manuscript. We thank the reviewer for spotting this aspect. Names in Table 1 and in the ms has been checked for consistency. Methods for measuring POC fluxes and the Total mass fluxes were stated in section 2.4, the text reads as follows: "Upon collection traps content was filtered through pre-combusted (450°C, 4h) and pre-weighed Whatmann GF/F filters. For total mass flux determination, filters were dried at 60 âŮęC for 24 h and re-weighed. For POC determination, filters were stored at -20°C until analysis, which was carried out by means of a Thermo Elementar Analyzer (Flash - EA 1112), after acidification with HCl for removing carbonates. The percentage of organic carbon on total mass (OC%) was calculated from POC fluxes and total mass fluxes."

L30, p6: if you decide to call the station outside the farm a "reference station" please use this terminology all through the manuscript. Thanks. We removed the term "reference", and opted for a more detailed description: " Early diagenesis processes underneath the farm and at a nearby station located outside the farm influence"

L33, p6: no information about how porosity was measured. Porosity was measured from the weight loss upon drying at 60°C. The first weighting was performed just after sample collection to minimize water loss. The water loss was then converted to porosity using sediment average dry bulk density and salt correction. A sentence was added in the revised version: "Porosity was obtained by measuring the weight loss upon drying until at 60°C until constant weight. Porosity was recalculated from this weight loss using salt correction and dry bulk density."

L36, p6: "Oxygen shows a quasi- monotonous decrease in concentration" you mean downward in the sediment? yes. This was specified in the ms.

L37, p6: what do you mean by low variability? thanks, the word "low" was replaced by the word "limited"

L4-5, p7: I clearly doubt that the trend in the only one DIC profile (and the two subsurface samples) reported can indicate anything about bioirrigation.

Thanks. We would like to remark that two sets of profiles indicating the same type of distribution were available: DIC and NH4, moreover the effects could be coroborated by the shape of the profile of another chemical species (SO42-), available from the same cores. This was stated in the ms (page 9 line 15, previous ms version):"The shape of the DIC–NH4+ profiles indicates bio-irrigation (Meile et al, 2001; Canavan et al., 2006), although the deeper increase is not visible in our data profiles due to limited penetration of the cores. Indeed, the very limited increase in concentration profiles in the first centimeters can only be linked to input of bottom water with lower DIC and NH4+ by irrigation, given the large recycling intensity in surface sediments as exemplified by O2 profile."

Based on the important reviewer's comment (see also general comment #2, and comment L13-14, p7, below), we included in the new version of the ms additional information on potential bio-irrigators. Details are available above, in the reply to general comment #2, here we report the text which was included in the new version of the ms, pg 9 : ". We remark here that $\alpha 0$ and xirr1 were the only two parameters calibrated at IN1, and they suggest a higher infauna activity, shifted towards the surface at this site. This feature was independently confirmed by a set of macrobenthos samples collected at the two stations as a part of a complementary study (Colla, 2017). Macrobenthos samples showed a higher diversity (48 vs 31 taxa recorded) and abundance (on average 1900 vs 1000 ind. m-2) at IN1 with respect to EST1, accompanied by the presence of larger organisms (0.065 g ind.-1 at IN1 versus 0.034 g ind.-1 at EST1). This is in

agreement with the expected influence of biodeposition from mussel culture (McKindsey et al., 2011). Species recognized as important bioturbators (as Lagis koreni, Glycera unicornis, Sipunculus nudus, Eunice vittata, Hilbigneris gracilis, Amphiura chiajei, Ensis minor, Dosinia lupinus, Tellina distorta, Nassarius incrassatus) were present in both samples, accounting for approximately 18% of the total abundance at EST1 and for the 35% at IN1."

L7, p7: $O_2$ with 2 in index (same all through the manuscript) corrected

L13-14, p7: Is the bioirrigation expected to decrease the porewater concentrations at a precise depth? I think there is a need to better explain how bioirrigation is taking into account in the model. Usually, the bioirrigation rate is applied over a depth interval with the intensity decreasing (linearly or not depending the model) with depth from a maximum value at the surface to a 0 value at a specific depth. This induces a dilution effect of the porewater by the overlying water decreasing with depth. Is this not the case here? If not, please precise this aspect. Thanks, this was the case, although the formulation used was a double exponential (see Table 2), which allowed to represent a sharp decrease of bioirrigation from its topmost value to 0. A similar shape, although obtained by adopting a different, discontinuous, function was reported by Canavan et al. (2006). We fully agree with the reviewer about the lack of details with respect to this point on the previous version of the ms. In the new one the following elements were added: - The formulation used to define the profile of the bioirrigation rate has now been added in Table 2 (formulation for bioturbation was also specified in Table 2); - A sentence pointing to the methodology used to include bioirrigation in the BRNS tool, used for developing the EDM was added in the methods section (ln 7-11, pg 4); - A plot showing the shape of bioirrigation rates in the two profiles was included in the appendix (Appendix, Figure A3)

In addition, information on the macrofauna species and bioturbation behavior may help to define the way to account for bioturbation (sediment reworking + bioirrigation) processes (in Colla, 2017?). Thanks for this comment, with respect to this point please

see our reply in the point above (L4-5, p7). We added information on this at pg 9.

L14, p7: I'm not agree with this affirmation. The range of concentration between measured and modeled profiles are similar but the vertical trends clearly differ. The sentence was rephrased, and now reads as follows: "In general, simulated profiles reasonably agree with observed concentration values, although differences between model and data vertical trends are visible, in particular at station EST1, were predicted $NH_4^+$ exceeds observed concentrations."

L17-20, p7: Figure 7 seems to show the diffusive $O_2$ fluxes for IN and EST stations. I can't see any comparison between modeled and measured data. Modeled data were represented by the black dot, in the new version we propose to use a more distinguishable marker (see new Figure 7).

L17-20, p7: how the measured ex situ $O_2$ profiles, and assessed $O_2$ diffusive fluxes, may be used here to gain information on the oxygen uptake in situ since: 1) profiles seems to haven't been performed at in situ temperature and in situ oxygen content, and 2) under sunlight influence that could be clearly higher than under in situ condition (observation microphytobenthos production that is know to strongly impact the diffusive fluxes at the sediment-water interface as well as the $O_2$ penetration depth). This is a crucial step that may to be clearly addressed. -

With respect to this important point, please see above our replies to the general point #3, and to the following specific aspects: L 21, p5; L35-36, p5.

L22-23,p7: the three pools of OM: you actually mean the total organic matter that correspond to 2 pools in EST station and 3 pools in IN station? yes, we rephrased the sentence as follows: " Figure 8 shows the partitioning among mineralization pathways, indicated by the electron acceptors, of the total organic matter."

L24-25: this is not what is reported in Figure 7. Thanks, the correct reference here is Figure 8.

Discussion

L24, p8: What is ED? It should be EDM as previously. For consistency, we used EDM throughout the ms.

L24-32, p8: You said that modeled POC flux from EDM agrees with POC flux measurements from a sediment trap. (that already was the subject of the previous section) but then try to explain why there is a factor 2 difference. We attempted to explain which we think could be the reasons behind this difference at lines 34-39, pg 8 (previous version). Based on this comment from the referee, we decided to change the introductory sentence, and now the text reads as follows: " Absolute values of the POC fluxes obtained from the sediment trap experiments can be cross-compared to the values estimated through the inverse use of the EDM. The value of 11.6 mmol C m-2 d-1, at EST1 accounts for approximately 50 % of the flux measured from the traps (22 mmol C m-2 d-1 on average). This difference can be primarily ..."

L12-13, p9: This is an important point.

Table 1: - title do not correspond to what can be seen in the Table - please add a mention specifying the stations that are inside the influence of the farm and those outside the farm (reference?) and gather results for stations IN and EST to help the reader. - be careful to the significant digits. - homogenize the position of the text inside the cell Thanks. Suggested changes have been implemented both on the Table, and on Table caption.

Table 2: - Second line corresponds to POC deposition I guess. This should be clearly mentioned. - please add a mention specifying the data for inside the influence of the farm and this outside the farm (reference?) to help the reader. - Thanks. Suggested changes have been implemented How can you assume that the CNP ratios are the same in both stations. Based on the close location of the two stations we considered reasonable to assume homogeneous CNP ratio for the background fluxes (OM1, OM2), which was based on available field data for this area (Brigolin et al., 2009) - no field

[Figure]

data were available to distinguish OM1 and OM2 - Due to the temporal variability of faeces to pseudofaeces ratio, and the temporal variability of CNP for each of these two types of biodeposits (both included in the OM3 pool), we considered difficult to estimate reliably the average CNP composition, and therefore proceeded with the conservative assumption of maintaining the CNP ratio of OM3 equal to the one of the remaining pools.

- mineralization rates reported correspond to total (OM1 + OM2 + OM3) organic carbon? yes, this was specified in the table in the new version of the ms

Figure 2: The caption is clearly no detailed enough? What are the red bubbles? The black rectangles? The blue lines? Thanks. We rewrote the caption, detailing the different parts of this cartoon figure.

Figure 4: Please add the simulation day on the figure. It will help the reader to follow the writing. This change was implemented.

Figure 5: - specify "por" in the caption or write "porosity" on the figure. Same for DIC. - I doubt the unity of porosity is % as mentioned. - the term "micro-profile" is only applicable to O2 and eventually to porosity profiles. Thanks, both figure and captions were changed based on the reviewer's suggestions.

Figure 6: precise this caption. Thanks, we extended the description in the caption.

Figure 7: - caption of the figure 7 is unrelated to the figure 7 - write on the figure (noted Figure 7) that it correspond to O2 diffusive fluxes (if this is well the case). Thanks, caption and figure were both modified.

Please also note the supplement to this comment:
https://www.biogeosciences-discuss.net/bg-2017-206/bg-2017-206-AC1-supplement.pdf

**Supplement:**

[revised manuscript text omitted]

5 **Figure 3**

[Figure]

**Figure 4**

[Figure]

**Figure 5**

[Figure]

**Figure 6**

[Figure]

**Figure 7**

[Figure]

**Figure 8**

---

## Author Comment (AC2) · 27 Jul 2017

General Comments First let me apologies for the late submission of these comments as I lose the original document with the formal comments. This is a quick summary of what I remember by going through the draft again. Additionally let me add to this that my experience is with physical marine numerical modelling and data analysis, and I am not particularly familiar with the application on mussel farming. These aspect will be noticed by the focus of my comments. The approached ideas and objectives are interesting, however I see issues (disagree) with some of the method and ways that

the data was used. This could be as I am unfamiliar to this particular application and a good answer could quickly clarify these aspects (see specific comments). At this point I will not recommend to publish this work if not further improvements are completed. It was difficult to follow the document at the beginning (until section 2.3), after this it was easier to follow the structure. However improving figures 1 and 2 will help to create a clear big picture at the start (please, see specific comments). The specific comments and technical corrections start with the page (P) number and line (L) number to indicate the specific place in the text that is commented.

We thank the reviewer for the useful comments. Our answer to specific points are reported below. We remark that in order to improve the readability of the work we followed the suggestions by editing figures 1 and 2, and largely re-writing captions of these figures. Based on a specific point made by referee #1 we enlarged the description of the work steps in the last part of the introduction, and removed the bullet point description present in the first version of the ms at pg 2 lns 30-40. We tried to track within this document most of the changes performed on the text, when this was not possible, changes were implemented only on the manuscript (ms), and reference reported in the present document. Reference to the initial submission and to the updated version of the manuscript were detailed. When not stated, we imply reference to the revised version. The revised version of the ms is provided as a supplement.

Specific Comments P4 L38: How reliable is the Chla and SST data when the study area is near the coast (at âĹij 2km and the data resolution is 4km)? We agree with the reviewer on the importance of this aspect. First, we would like to remark that the simulation predicting growth and faeces and pseudofaeces production at the farm throughout the year was aimed at characterizing the behavior of a typical farm in this area during one decade (2002-2012).

In the present work we adopted two precautions aimed at smoothing possible artificial variability of Chla and SST data originated by the use of earth observation algorithms at coast, from which, in river influenced coastal areas, one could expect an overestimation

of Chla concentration induced by confounding non-phytoplanktonic color particulate matter. These were: 1) data were averaged over spatial domain wider than the area covered by the mussel farm (10km x10km=100 km2 versus 4 km2 of the farm - 2x2 pixels from satellite images included in the area); 2) 10 different runs were performed with the mussel growth model (population), and only the median trajectories (for faeces and pseufofaeces) were considered in the following deposition and early diagenesis model.

Therefore, one possible issue could have been the excessive smoothing of chlorophyll-a peaks ( blooms), inducing an under-estimation of mussel growth. However, in terms of growth trajectories the model reported results which are comparable two growth patterns previously observed for mussels in this area (a comment about this is present in the results, section 3.1).

A further important point to mention is that similar approaches, based on satellite derived chlorophyll-a concentrations (Chl-a) and sea surface temperature data (SST) were reported for modeling the growth of the mussels in coastal areas (e.g. Filgueira et al., 2013; Thomas et al. 2011). In terms of growth and metabolic rates predictions, also these shellfish models were able to perform adequately, although being forced with time series of satellite data.

Y. Thomas, J. Mazuri, M. Alunno-Bruscia, C. Bacher, J.-F. Bouget, F. Gohin, S. Pouvreau, C. Struski. Modelling spatio-temporal variability of Mytilus edulis (L.) growth by forcing a dynamic energy budget model with satellite-derived environmental data J. Sea Res., 66 (2011), pp. 308–317

Filgueira, R., Comeau, L.A., Landry, T., Grant, J., Guyondet, T., Mallet, A.: Bivalve condition index as an indicator of aquaculture intensity: A meta-analysis. Ecol. Indic., 25, 215-229, 2013.

If you imply that the average of certain number of Chl-a and SST data points is representative of the study area, Where is explain in the methodology? Thanks, lat and long

edges of the domain in which data were collected were reported in the ms. Based on the reviewer comment we decided to add a sentence in the new version of the ms, in order to make the process more clear: "Chlorophyll-a and SST data were derived from the sensor Modis (Moderate Resolution Imaging Spectroradiometer) Aqua and Terra respectively, with a spatial resolution of 4km. Data were spatially averaged over the area defined above. "

For the coordinates mentioned in P4 L37 the number can be guessed, but it will be better to see the number. And finally, Is it representative for the 2km2 study area the Chla and SST time series if they were constructed based on an area of app. 100Km2?

Please see our reply to the two previous comments.

P4 L38: How do you address the problem of color particulate matter, optical shallow waters and water turbidity, where the river could have a major role to play? As example please see: * Cannizzaro and Carder (2006)â ÌÍA ÌĘ a [optical shallow waters]

Thanks, again, we believe that this is related with the use that was done of Chla data, which were provided as input to characterize the typical growth trajectory of a mussel over one decade, and in order to assess the mean Delta in POC flux induced by the presence of the farm. Based on this comment from the reviewer, we decided to add a specific sentence regarding the perspective use of the model (section 4.3 "Integrated model features"). The sentence reads as follows: "We underline that the application presented in this work could be extended, in order to include the evaluation of the uncertainties related to spatial inconsistencies of nearshore-offshore remote sensing products."

P5 L6: Could you show the current meter data? as you say that you are using the residual current edited. Why did you add storms? How many storms did you add in this random process? (You say that this method should be preferred for forecasting, however you are reproducing a period where you have data to validate and calibrate, Is not this closer to a hind-casting than a forecasting). Why you are not using the current

due to tides? Tidal currents will have an impact on particles that represent the faeces and pseudofaeces

Thanks. Based on the reviewer's comment, lines 5-13 at pg 5 of the previous draft were rewritten. A compass plot showing the time series of currents data was added to the supporting materials (Figure A2).

The number of storms was added on the basis of the incidence of storms detected within the time series analyzed (for the months from March to September), which was integrated by the analysis of current meter data deployed nearby the study area within previous studies (Rampazzo et al., 2013; Giovanardi et al., 2003 – in Italian). These latter data refer to year 2006 (July-September) and 2007 (April-May), and years 2001-2002 (from October 2000 until August 2002). With respect to the currents time series analysis, we extracted the residual current and the slowly varying tidal components, in order to separate the short-period fluctuations related to turbulence, which were accounted in the model by random displacement. We followed the procedure as described in the paper by Jusup et al. (2007), which was the source publication for our deposition model. For completeness, we report that a subsequent validation of the algorithm proposed by Jusup et al. (2007) in the Adriatic Sea was reported by Jusup et al. (2009), and therefore the model was coupled with a model of the mussel farm population in Brigolin et al. (2014).

We removed the reference to hind-casting and forecasting, since we think that this could generate confusion in the reader. With respect to the referee's question, we believe that the experiment performed in this work is closer to a forecasting simulation, since: - Currents are not provided directly from the current meter; - Sediment traps used for comparing model outputs (corroboration) were not deployed at the same time than current meter, and are therefore used to compare the typical deposition of the farm, and not with the aim of performing a strict model validation experiment. We believe that this type of simulation is in line with the other methodological choices previously explained, aimed at assessing the "typical" behavior of the farm, in terms of

deposition.

With respect to this point, the text included in the new version of the manuscript is the following: "Modelling deposition requires an input time series of water velocity at an hourly time step. These data were provided on the basis of a current meter deployment carried out between March and September 2010 at a station located approximately 500 m from the NE edge of the farm (Boldrin A. pers. comm). Current meter data were first processed by means of a classical harmonic analysis, in order to extract tidal components as well as long-term residual means (Pawlowicz et al., 2002). On the basis of the procedure proposed by Jusup et al. (2007), the residual currents were therefore edited randomly for short periods of time in order to reproduce the variability recorded from current meter measurements during extreme events (i.e. storms). Number of events was imposed on the basis of the 2010 current time series, and of previous current meter deployments available for this area (Rampazzo et al., 2013; Giovanardi et al., 2003). Effects of tide and storm events were therefore accounted in the final time series, while short-period fluctuations related to turbulence were accounted for by the deposition model, as reported by Jusup et al. (2007)."

Rampazzo, F., Berto, D., Giani, M., Brigolin, D., Covelli, S., Cacciatore, F., Boscolo, R., Bellucci, L. G., Pastres, R.: Impact of mussel farm biodeposition on sediment bio-geochemistry in the north-west Adriatic Sea. Estuar., Coast. Shelf S., 129, 49-58, 2013.

Giovanardi, O., Cornello, M., Tiozzo, K., Casale, M., Franceschini, G., 2003. Effetti degli aggregati mucillaginosi sulle comunità macrozoobentoniche al largo di Chioggia. Programma di monitoraggio e studio sui Processi di formazione delle Mucillagini nell'Adriatico e nel Tirreno (MAT) Rapporto finale vol. II. ICRAM, Roma, pp. 351-366.

P6 L11: Please explain further in the method section that the mean of Chl-a and SST is spatial and temporal, resulting in a mean or characteristic year for each variable. This means that strong events of Chla are smooth as shows figure A1b, which means

implication for the model individual growth and population model.

Thanks. As reported above, the sentence – data were spatially averaged over the area defined above – was added, bringing to the following specification: "Chlorophyll-a and SST data were derived from the sensor Modis (Moderate Resolution Imaging Spectroradiometer) Aqua and Terra respectively, with a spatial resolution of 4km. Data were spatially averaged over the area defined above.". As far as the temporal average is concerned, this was not performed on SST and Chl-a data, and one different model trajectory was provided for each simulated year (see plates in Figure 3). Results of the mussel population growth model were therefore summarized as a median value in order to provide an average input for the deposition model (this step is specified at lines 23-24 pg 5 "The median daily value of faeces and pseudofaeces fluxes from the 10 simulations was used as an input for the deposition model.". We also attempted to resume this within the cartoon presented in Figure 2. With respect to the influence that this could have on the growth model, please see our reply to comment P4 L38 above.

P6 L34: If you are going to use cm here, then please use cm in the figure. Thanks. Units were uniformed to mm for porosity.

P7 L36: Are the features and methods of Weise et al, (2009) comparable to yours? Depth of the study area, mean velocity (if they are considering residual current or not), size of the farm, etc The same comment goes of the Hatstein and Steven (2005) reference.

Thanks, as mentioned in the text, Hatstein and Steven (2005) and Weise et al (2009) were reported in our work as examples of previous studies focusing on modeling shell-fish deposition. Concerning methods, models are comparable, being all based on a Lagrangian approach, however, in both cases, POC originated from the farm was not modeled, but prescribed to the model a-priori. This aspect was underlined in the last section of this ms (section 4.3).

As regards bathymetries there are differences, 14 m in this work, 20 m in the exposed

site located in Cascapedia (Weise et al., 2009), 20 m in the more sheltered site considered by Hatstein and Stevens (2005), New Zealand. Our average current velocity was 5.4 cm s-1, in the exposed farm in Cascapedia 10 cm s-1, while in New Zealand the velocity was 3.4 cm s-1.

The comparison that was reported at the beginning of the discussion, was aimed at underlining this potential variability in the extent of the dispersal area, and the fact that our case study represents an intermediate between the other two reported. The sentence is the following - for the sake of clarity, we added details about our case study, depth and mean current velocity in brackets on the first line:

"The extent of the depositional area obtained in this study (on average 50 m from the edge of the farm; 14 m depth; mean current velocity of 5.4 cm s-1) can be compared with the results obtained in previous studies. In an exposed site, Weise et al. (2009) (Cascapedia Bay, Canada; 20 m depth; mean current velocity of 10 cm s-1), constrained the area of higher organic enrichment within 90 m from the edge of the farm. Dispersal area reported by Hatstein and Stevens (2005) was smaller, extending with a radius of approximately 30-40 m from the edge of the farm (20-30 m depth; mean current velocity of 3.4-4.0 cm s-1). These differences in extent of the dispersal areas seem to be primarily associated to the action of currents and wave energy inducing resuspension of biodeposits accumulated on the seabed (see Cromey et al., 2002)."

Figure 1 : The figure does not explain itself. It is not possible to know the meaning of the rectangle and and points in the figure. In my first look at the figure, the river was a road and I have to google the river to find out where it was.

Thanks. The caption was rewritten, including information on the different elements of the figure, and distinguishing station under the influence of the farm from nearby stations, located outside the influence of the farm (as suggested by reviewer #1). Colors of the river and the lagoon waters were also changed, in order to improve the figure readability.

Figure 2 : The figure caption is not good. Please give to the reader enough information for the "Information flow". Please add to the boxes in the diagram the reference to each represented/used model. Thanks for this useful suggestion. This was implemented it in the new version of the caption. Why are you using the word reanalysis for the current input in the particle tracking model (faeces and pseudofaeces)? Here wording was a mistake, and text was rephrased. "Reanalysis" did not refer to the procedures used by studies using data assimilation techniques.

Figure 3 : What is the meaning of the sharp changes shown by graph a? It will better If the caption has more information about the figure. As example, if this represented data are modeled results or measurements. Discontinuities shown by graph-a are due to weight loss after spawning events. These are all model results, and details about model theory are available on the paper by Brigolin et al. (2009). Based on this comment we added additional explanations to the figure caption.

Figure 5: Squares and triangles without legend. Y-axis without label just units. Thanks, the figure caption was rewritten, in order to clarify these aspects.

Figures in general: Improve caption for the figures to stand by themselves. Thanks. We worked on this aspect throughout figures and tables.

P6 L23: Does this agree with the mean current velocity, mean vertical velocity for the particles to sink and mean depth? considering the free fall of a particle, an average depth of 10m (particle released at the middle farm), and the fastest particle sinking velocities (1.0 cm s-1 for faeces), the 200m displacement means an average current velocity for that day of approximately 20 cm s-1, which is in agreement with the current meter observations during storm events (see fig A2).

Technical Corrections

P3 L27: Define acronym POC earlier. There are few times before this where the acronym is used. Thanks, in the new version of the ms POC is independently defined in the introduction and in the abstract. In the title, the acronym POC has been substituted by the extended name.

P4 L16: Reference Cappellen and Wang, 1996 is not listed in the references The reference is Van Cappellen and Wang, 1996. The name of the first author was reported with the capital V in the text and in the reference.

P5 L 26: Define the acronym PVC. done

P6 L6: Define acronym HPLC done

Boudreau (1996) and Sanchez-Jerez et al (2016) listed in the reference but I did not find them in the text un-cited references were removed.

Please also note the supplement to this comment:
https://www.biogeosciences-discuss.net/bg-2017-206/bg-2017-206-AC2-supplement.pdf

**Supplement:**

[revised manuscript text omitted]
 + (\frac{4x + 3y}{5})NO_3^{-} \xrightarrow{-R_2}$   $(\frac{2x + 4y}{5})N_2 + (\frac{x - 3y + 10z}{5})CO_2 + (\frac{4x + 3y - 10z}{5})HCO_3^{-} + zDIP + (\frac{3x + 6y + 10z}{5})H_2O$ 3. Sulphate reduction  $(CH_2O)_x(NH_3)_y(H_3PO_4)_z + (\frac{x}{2}) \cdot SO_4^{-2-} + (y - 2z)CO_2 + (y - 2z)H_2O \xrightarrow{-R_3}$   $\frac{x}{2}TS + (x + y - 2z) \cdot HCO_3^{-} + yNH_4^{+} + z DIP$ Secondary redox reactions 4. Nitrification  $NH_4^{+} + 2O_2 + 2HCO_3^{-} \xrightarrow{-R_4} NO_3^{-} + 2CO_2 + 3H_2O$ 5. Sulphide oxidation by  $O_2$  $TS + 2O_2 + 2HCO_3^{-} \xrightarrow{-R_5} SO_4^{-2+} + 2CO_2 + 2H_2O$

Table A3. Rate laws used in the EDM. Rates refer to the reactions listed in Table A2 of this Appendix.  $f_i$  were computed according to the formulation reported in Aguilera et al. (2005).

Rate laws

$$R_{1} = f_{o2} \cdot k_{OM_{k}} \cdot [OM_{k}] \cdot k_{acc} , with \ k=1,2,3$$

$$R_{2} = f_{NO_{3}^{-}} \cdot k_{OM_{k}} \cdot [OM_{k}] \cdot k_{acc} , with \ k=1,2,3$$

$$R_{3} = f_{SO_{4}} \cdot k_{OM_{k}} \cdot [OM_{k}] , with \ k=1,2,3$$

$$R_{4} = k_{4} \cdot [NH_{4}^{+}] \cdot [O_{2}]$$

$$R_{5} = k_{5} \cdot [TS] \cdot [O_{2}]$$

Table A4. Reaction specific and general parameters used in the EDM. (1=Wang & Van Cappellen, 1996; 2= Jourabchi et al., 2005;3= Canavan et al., 2006; 4= Berg et al., 2003)

| Parameter name                    | Value                        | Units                         | Description                                             | Source                     |
|-----------------------------------|------------------------------|-------------------------------|---------------------------------------------------------|----------------------------|
| O 2 lim                | $16.0 \times 10^{-6}$        | mol L -1           | limiting concentration for O 2               | Mean value from 1,2,3,4    |
| NO 3 - lim  | $4.7 	imes 10^{-6}$          | mol L -1           | limiting concentration for NO 3 - | Mean value from 1,2,3,4    |
| SO 4 2- lim | $1180.0\times10^{\text{-6}}$ | mol L -1           | limiting concentration for $SO_4^{2-}$                  | Mean value from 1,2,3      |
| $k_4$                             | $1.2 	imes 10^7$             | $(mol L^{-1})^{-1} y^{-1}$    | kinetic constant for Nitrification                      | Mean value from 1,2,3,4    |
| K 5                    | $2.7 	imes 10^8$             | $(mol L^{-1})^{-1} y^{-1}$    | kinetic constant for sulphides oxidation                | Mean value from 1,2,3,4    |
|                                   |                              |                               | by O 2 kinetic constant                      |                            |
|                                   | 0.25                         | $\mathrm{cm} \mathrm{y}^{-1}$ | Vertical velocity                                       | Alvisi and Frignani (1996) |
|                                   | 10 -4             | У                             | 
[revised manuscript text omitted]

---

## Author Response (AR1)

**Associate Editor Decision: Reconsider after major revisions** (06 Sep 2017) by Silvio Pantoja

Comments to the Author:

Review of bg-2017-206 by Brigolin et al.

*Dear Editor,*

*Our answers to the different questions are reported below in this document (text in Italics). Changes performed on the manuscript have been highlighted in yellow.*

*We would like to thank again you, and the two anonymous reviewers, for the detailed and useful comments made.*

*All the best,*

*Daniele Brigolin*

1) Please avoid Acronyms. The scope of this journal is an integrative, and the excessive use of abbreviations, specially non-standards only makes the article more difficult t o read, unless you come from this specific field.. AE, TSS, TSM (only used twice…), etc. *The following acronyms have been removed from the text: AE, TSS, TSM, SST.*

2) Reviewer 1. AC1, C6. L18-19, p4: What is the impact of neglecting Fe and Mn biogeochemical related processes on the total biogeochemical processes? Are these processes really negligible? This could be the case as sulfate reduction process appear negligeable (from $SO_4^{2-}$ profiles and in spite what your modeling result tend to indicate). In addition you do not have any $NO_3^-$ data that could also be an important process in the area studied. -

Response: Thanks. Including Mn and Fe related processes would definitely increase the EDM realism, but also its complexity. Mn and Fe dynamics were present in a previous modelling work applied in a close area, located in the Northern Adriatic (Brigolin et al., 2011, cited in the text). However, with respect to the present work, in this other case, a larger database was available for constraining the model (pg 4, lns 18-20 - initial version of the ms). This model limitation was stated explicitly in the methods section of the ms, and a remark concerning the potential development of the model was also present in the discussion (pg 9, lns 43-44 - initial version of the ms) " A more precise estimation of the fate of this oxygen could be obtained by introducing in the model FeS precipitation, for which at least $Fe^{2+}$ measurements in pore waters would be required.".

Editor: This question needs to be properly answered. We need to know the impact of neglecting: overestimate, underestimate, by how much, is it still reliable without , etc.

*It is known from the early work of Jorgensen (1982) that sulfate reduction is a very efficient mechanism of organic matter mineralization in coastal sediments and represents 50% or more of total mineralization .*

*Metal reduction has been shown to play a role in metal-oxides rich sediments (Canfield et al., 1993), but is believed to be of less importance in most settings (Burdige, 2006), although the debate is ongoing (Beckler*

*et al., 2016). There are very few quantifications of the relative role of metal oxides and sulfate reduction for early diagenesis (and none for the Adriatic). In their 1996 paper built on Canfield's result of 1993, Van Cappellen and Wang (1996) showed that the OM mineralization by SO4 corresponded to 60% of total mineralization whereas Fe and Mn oxides reduction corresponded to 20%. Based on data reported in Burdige (2006), in shallow depth environments, Fe and Mn contribute on average for 10% of the total mineralization, with peaks around 20%. From these calculations, we can express that mineralization rates calculated without iron and manganese oxide reduction, as is the case for this paper, may underestimate the total OM mineralization by 20%, but are still reliable for the most part.*

Response: With respect to NO3, we acknowledge that this would be a relevant information for constraining nitrification/denitrification, and this was highlighted in the discussion (pg 10, lns 5-9 - initial version of the ms) " Higher NH4+ concentration predicted at station EST1 with respect to field data could be explained with a higher rate of nitrification at this station. However, in the calibration performed within this work, the kinetic constant for nitrification was kept at its original value (Table A4), due to the lack of data concerning NO3-."

Editor: Again, What is the impact…? (please see above)

*The role of denitrification is generally minor (a few percent of the total) in OM mineralization. This is the case in Canfield's paper in the North Sea (1993; 3-6%), in the Mediterranean near the Rhone River (Pastor et al., 2011; 0.1-4%), in the Black Sea (Capet et al., 2016; 5-6%) and in the Northern Adriatic (Capet, unpublished data; 2-4%). Our model results indicated that denitrification contributed between 1 and 2% to the total mineralization. Based on the comparison reported above, we can say that the contribution of denitrification to the total mineralization of organic matter could have been underestimated, due to the lack of NO3 data, of 1-2%.*

3) Reviewer 1: C7 Last paragraph. L5-6, p5: what is the influence of the farm on the current within the farm? This is the current within the farm that will drive the transport and deposition of particulate from the farm. This is not explained within the deposition model section. I think this is important for the reader. Please explain.

Response: This engineering aspect, although of interest, has not been properly investigated yet*. To the best of our knowledge, multiple current meter measurements within the same longline suspended mussel farm are not available, and model predictions regarding the local effect of structures on currents would be complex to validate. Similar effects are currently neglected in most deposition models also for fish cages (which nets are expected to have more relevant impacts on the current flows). We added a synthetic remark on the fact that the aspect is not included in the model (model theory paragraph 2.2, ln 35 pg 3).

Editor: *There is information in the literature. See for instance:

Deposition beneath long-line mussel farms

([http://www.sciencedirect.com/science/article/pii/S0144860905000166](http://www.sciencedirect.com/science/article/pii/S0144860905000166))

*Thanks, we clarify the following:*

*This aspect has not yet been properly investigated for the type of structures used in the Northern Adriatic (traditional longlines with suspended ropes). In their work Hartstein and Stevens (2005) assessed the effect on currents of New Zealand-type mussel long-lines, by means of 2D acoustic current meter measurements.*

*The structure of New Zealand long-lines is markedly different from the traditional static long-lines, used also in the Adriatic Sea. From the engineering point of view, the first ones are based on a continuous cable, while the second one on a discrete number of suspended ropes. This difference was underlined also in the work by Hartstein and Stevens (2005), pg. 195 and 198. A sentence on the fact that the aspects concerning the interaction between the farm structure and the water current is not included in the model, is present in the manuscript (model theory paragraph 2.2, ln 35 pg 3).*

4) Reviewer: AC2, C4, Fifth paragraph. P4 L38: How do you address the problem of color particulate matter, optical shallow waters and water turbidity, where the river could have a major role to play? As example please see: * Cannizzaro and Carder (2006)â ǀlA ǀĘ a [optical shallow waters]

Response: Thanks, again, we believe that this is related with the use that was done of Chlorophyll-a data, which were provided as input to characterize the typical growth trajectory of a mussel over one decade, and in order to assess the mean Delta in POC flux induced by the presence of the farm. Based on this comment from the reviewer, we decided to add a specific sentence regarding the perspective use of the model (section 4.3 "Integrated model features"). The sentence reads as follows: "We underline that the application presented in this work could be extended, in order to include the evaluation of the uncertainties related to spatial inconsistencies of nearshore-offshore remote sensing products."

Editor: I do not understand your answer. Please rephrase and address the problem

*Northern Adriatic optically shallow waters are influenced by river discharge, and estimations of chlorophyll-a (CHL-a) concentrations could be potentially over-estimated due to the interference caused by Colored Dissolved Organic Matter absorbance, and bottom reflectance (see Cannizzaro and Carder, 2006). However, two main considerations drove us to use Modis CHL-a data provided by the EMIS monitoring service:*
1) *according to Mauri et al. (2007), the area investigated in the present work lies at the edge of the zone under Po river influence (the main river in the Northern Adriatic sea) and the local river does not provide perturbation of water clarity except during major floods (avg runoff of the Sile river is approximately 1 order of magnitude lower than Adige and Brenta, and 2 orders lower than Po, see text pg. 8 ln 30 and the data reported in the work by Cozzi and Giani (2011));*
2) *in the present work, chlorophyll-a concentration estimated from satellite data were used to force the mussel growth model, in terms of the increase in soft tissues dry weight, and in the length of the shell, which were found to be comparable to the field observations for this area (see remark at pg 6, line 29). This was regarded as a first indication of the fact that model results were not biased by an overestimation of CHL-a concentrations. Furthermore, in our analysis, the mussel growth model was run for 10 years (from 2002 until 2011) with correspondent Satellite Chla fields, providing for each single year an independent output for the growth (weight-length) and the faeces-pseudofaeces (reported in Figure 3). The average Delta in POC flux induced by the presence of the farm was quantified on the basis of the median value of the 10 years of model results (dotted lines in Figure*

*3). This procedure guaranteed a further smoothing of eventual peaks induced by an over estimation of CHL-a due to the use of non-optimal algorithms for ocean-color processing in coastal waters.*

*Mauri et al. MODIS chlorophyll variability in the northern Adriatic Sea and relationship with forcing parameters. JOURNAL OF GEOPHYSICAL RESEARCH, VOL. 112, C03S11, doi:10.1029/2006JC003545, 2007*

5) Page 2, Line 7. This sentence distracts, and does not add anything new to the introduction and its flow. "Biogeochemical and contaminant cycling associated to the presence of shellfish farming in a Mediterranean lagoon was the focus of a special issue (Estuarine Coastal and Shelf Science 72(3); Rabouille et al., 2007). " *The sentence was removed.*

6) It is not clear if "…representative for the 2km2 study area the Chla and SST time series if they were constructed based on an area of app. 100Km2? " (Reviewer 2), considering that in line 25, Page 3, is stated "…The individual growth model requires in input time series of daily values of sea water temperature, and concentrations of chlorophyll-a, particulate …"

Editor: This concerns needs a better explanation since averaged data (100 km2) is much bigger than farms and there several issues related to coastal calibration of chlorophyll, including reflection in shallow water (<40 m depth)

*CHL-a and SST time series were used to provide an input to the mussel growth model, from which we estimated the average Delta in POC flux induced by the presence of the farm. This involved the realization of 10 different model runs with associated forcings (Satellite Chla and STT), which were performed with the mussel growth model (population) along 10 years. Given the location of the farm (please see fig below), and considering that within this shallow area the bathymetry does not present drastic changes, we considered more representative to take the concentration as an average among the 4 pixels adjacent to the farm, rather than considering the value for one single pixel. In addition, this operation allowed: 1) to smooth eventual noise on CHL-a concentration induced by river influence; ii) to cover the whole time-window, which using a single pixel would have been limited by the presence of missing data due to cloud coverage.*

[Figure]

*Figure showing farm location (red rectangle) with respect to the pixels providing the resolution of satellite data used in this work.*

7) Other issues:

a) Fix "Chloophyll-  " in Fig. 2 *Figure Edited*

b) Proposed conclusion:

"to characterize more strongly". Do you mean, "better characterize"? *Edited*

Those issues need to be addressed for publication in the journal.

Sincerely yours

Silvio Pantoja

Associate Editor

[revised manuscript text omitted]

---

## Referee Report (RR1)

Dear Dr. Brigolin,

Thanks for replying the comments on your article Modelling biogeochemical processes in sediments from the north western Adriatic Sea: response to enhanced POC fluxes. The answers to the editor's observations address most of the aspects commented by the reviewers. It would have been nice to see the same response quality for the reviewers' comments.

Let me comment on your answer to the editor's observations related to using remote sensing products (Sea Surface Temperature and Chlorophyll-a concentration ).

The fact that your model represents the evolution of the mussels' features to values similar to the ones observed within the farm, it is used to indicate that values of Chl-a and SST from remote sensing products represent the study area. This is a good point, but if someone shows that the Chl-a o SST values used by your model have issues, then questions are raised about the model and how did you use it. Which means, this results as the only support it is not good.

As example, a quick look into Chla MODIS data shows that your study area is located in a zone with Chl-a values greater than 1.5-2 mg/m$^3$ most of the year. This feature points that the coastal waters of your study area are under the influence of "something" that enhances the Chl-a concentration or its estimation is affected on remote sensing products. You have not address this point and this questions your model results!!!. It looks like rivers could have a role/impact in the study area. This need to be supported in a better way, to be in the edge means that the river could have a significant influence considering that you use the four pixel average.

Some ideas that could help you to address the aforementioned comments at some level.

1.  Observation of Chl-a and SST that could validate the values of Chl-a concentration and SST are not over- or under-estimated and support that values are greater than 1.5-2 mg/m3 most of the year. Lavigne et al. (2015) describe some observations near your study area. https://www.biogeosciences.net/12/5021/2015/
2.  To state the limitations of using Chl-a data based on remote sensing products and make a sensitivity analysis of the model. As example, to describe a variability range for Chla values where the final modeled results are still similar to the observed with the mussel farm.

Another comment about using Chla and SST remote sensing products in your model. These products are surface observation and mussels are located deeper than 2-4 meters without considering that the ropes are another 4 meters suspended downward. Not sure if the Chl-a concentration is affected at that depth, but I am more concern about the values of temperature. The optical properties of the water in the study area will have a significant role in the vertical variability, as well as the riverine water could impart the SST temperature. Surface riverine water is usually a thin surface layer that could lead your model to be using under-estimation of temperature.

One last point, I did not have the time to look into the model used for representing the mussel growth, but the model could be a solution itself if the remote sensing values differ from observations. As example, a linear model can be calibrated to reach the values observed by changing the linear coefficient. If this was the case, this could complement the point 2. To state that limitations of using Chl-a and SST and how the method is planned to address this issue by calibrating the linear coefficient (if this is your case, probably not)

---

## Author Response (AR2)

January 16, 2018

Dear Editor,

Please find below in this document our reply to your specific comments, and revisions carried out on the manuscript in track-changes mode.

Thanks again for your comments on this manuscript.

All the best

Daniele Brigolin

Associate Editor Decision: Publish subject to minor revisions (review by editor) (09 Jan 2018) by Silvio Pantoja

Comments to the Author:

January 8, 2018

Dear Dr. Brigolin,

Thanks for your responses, that I found not satisfactory and require further clarification:

1.      "The median daily value of faeces and pseudofaeces fluxes from the 10 simulations was used as an input for the deposition model. This step also had the effect of smoothing potential biases in chlorophyll-a estimation, induced by the interference caused by Colored Dissolved Organic Matter absorbance (see Cannizzaro and Carder, 2006)."

Please summarize the argument of Cannizzaro and Carder that diminishes interference with CDOM

Thanks. With respect to these sentences we clarify the following:

1) the step which had the effect of smoothing potential biases induced by an over-estimation of chlorophyll-a concentrations is the computation of the median value over the 10 years.
2) the work by Cannizzaro and Carder (2006) was cited for providing an introduction on processes biasing chlorophyll-a estimation from remote-sensing reflectance in optically shallow waters, and does not bring arguments about methodologies for diminishing interference with CDOM.

Based on the editor comment, we decided to rephrase the text, in order to increase its clarity. We also referred to the work by Sarà et al. (2013), in which a similar approach was adopted for forcing individual-based models with satellite-borne data. This text in section 2.3 now reads as follows:

*"The model application was aimed at constraining the typical variability of mussel deposition at the farm site. In order to simulate the average flux of POC deposited by the farm, we carried out 10 different runs, considering each one a rearing cycle under forcings for a different year within the 2002-2011 time frame. Satellite data were used as inputs for individual-based population dynamics model, in accordance with previous studies (e.g. Thomas et al., 2011; Filgueira et al., 2013). The median daily value of faeces and pseudofaeces fluxes from the 10 simulations was used as an input for the deposition model. In computing this statistics we followed an approach similar to the one reported by Sarà et al. (2013), in order to smooth potential biases introduced in the model through the forcing data. This precaution was adopted since Northern Adriatic optically shallow waters are influenced by river discharge, and chlorophyll-a concentrations could be potentially over-estimated in specific months due to the interference caused by Colored Dissolved Organic Matter absorbance (Cannizzaro and Carder, 2006)."*

2.      One of the reviewers argued that satellite chlorophyll data was not well sustained. Your reply with supporting evidence should be in the text including reference "…according to Mauri et al. (2007), the area investigated in the present work lies at the edge of the zone under Po river…. ". Reviewer also suggest that you revise "Lavigne et al. (2015) describe some observations near your study area. https://www.biogeosciences.net/12/5021/2015/ …" and incorporate in the text. Do data match?

Thanks. We included supporting evidence in the manuscript. In addition, we read with interest Lavigne et al. (2015). Unfortunately, as reported in section 2.1 of their paper, the fluorescence data used to derive the statistics on chlorophyll-a concentrations were selected only in areas which exceeded the 100m depth, and thus not included the coastal area under study. In order to comply with the useful suggestion by the reviewer (i.e. to compare our satellite-derived chlorophyll-a concentrations with in-situ data) we therefore selected the work by Solidoro et al. (2009) in JGR.

The text now reds as follows:

*"..Our choice to rely on satellite-derived chlorophyll-a concentrations was supported by two main considerations:*

*- the analysis by Mauri et al. (2007) reported very weak correlations between Po River discharge and interpolated satellite derived chlorophyll-a concentrations in the area interested by this study, and the Sile river does not provide perturbation of water clarity except during major floods (average runoff of the Sile river is approximately 1 order of magnitude lower than Adige and Brenta, and 2 orders lower than Po (Cozzi and Giani, 2011));*

*- the median values of chlorophyll-a concentrations obtained from satellite data (see description below, and Figure A1) were compatible with the median values reported by Solidoro et al. (2009) based on the analysis of a 20 years of data (1986-2006) of in-situ chlorophyll-a measurements, on a portion of sea which included our study area (referred as sector 3 in their work), ranging between 1.35 and 2.38 □g L-1 in the upper layer (0-7.5 m).*

*Time series of monthly sea surface temperature and concentration of chlorophyll-a were extracted from the EMIS (http://emis.jrc.ec.europa.eu/) data base from July 2002 to December 2012, (longmin 12.5; longmax 12.6; latmin 45.4; latmax 45.5) by means of the R package EMISR v0.1 (R version 3.0.3). Chlorophyll-a concentrations and sea surface temperature data were derived from the sensor Modis (Moderate Resolution Imaging Spectroradiometer) Aqua and Terra respectively, with a spatial resolution of 4km. Being the farm located at the intersection of 4 pixels, in an area characterized by a flat bathymetry, it was considered more representative to take the concentration as an average among the 4 pixels, rather than selecting a single value. Due to the lack of long term time series of data, an average POC concentration had to be imposed, 0.1 mg L-1, on the basis of a time series of monthly data collected at a nearby farm between 2006 and 2007 (Brigolin et al., 2009). The Particulate Organic Matter / Total Suspended Solids ratio was fixed on the basis of the time series collected within the same work (Brigolin et al., 2009), providing an average Absorption Efficiency of 0.6."*

Minor issues:

3. Delete word "see" in Cannizzaro and Carder, 2006 and in the rest of the text; not needed.

Thanks, the word see was removed from citations.

1. Abstract, line 6 "…linked to new data in…". What is new data?

"…linked to new data in…" was rephrased and now reads as follow *"OM degradation pathways were investigated by constraining an early diagenesis model by using original data collected in sediment porewaters"*

2. Abstract. "… by similar proportions between oxic and anoxic degradation pathways at the two stations… ."

Change "degradation pathways" to "degradation rates"

Thanks. The suggested change was implemented.

Sincerely yours

Silvio Pantoja

Associate editor

[revised manuscript text omitted]

Figure 2

[Figure]

[Figure]

[Figure]

[Figure]

**Figure 3**

[Figure]

**Figure 4**

[Figure]

**Figure 5**

[Figure]

**Figure 6**

[Figure]

**Figure 7**

[Figure]